# Effects of SO₂ on optical properties of secondary organic aerosol generated from photooxidation of toluene under different relative humidity

Wenyu Zhang[1,2], Weigang Wang[*,1,2], Junling Li[1,2], Chao Peng[1,2,b], Kun Li[4,a], Li Zhou[5], Bo Shi[1,2], Yan Chen[1,2], Mingyuan Liu[1,2], and Maofa Ge[*,1,2,3]

[1] State Key Laboratory for Structural Chemistry of Unstable and Stable Species, Beijing National Laboratory for Molecular Sciences (BNLMS), CAS Research/Education Center for Excellence in Molecular Sciences, Institute of Chemistry, Chinese Academy of Sciences, Beijing 100190, PR China

[2] University of Chinese Academy of Sciences, Beijing 100049, P. R. China

[3] Center for Excellence in Regional Atmospheric Environment, Institute of Urban Environment, Chinese Academy of Sciences, Xiamen 361021, PR China

[4] Air Quality Research Division, Environment and Climate Change Canada, Toronto, Ontario M3H 5T4, Canada

[5] College of Architecture and Environment, Sichuan University, Chengdu, China

[*] E-mail addresses: W. Wang (wangwg@iccas.ac.cn) and M. Ge (gemaofa@iccas.ac.cn)

[a] now at: Laboratory of Atmospheric Chemistry, Paul Scherrer Institute (PSI), 5232 Villigen, Switzerland

[b] now at: State Key Laboratory of Organic Geochemistry, Guangzhou Institute of Geochemistry, Chinese Academy of Sciences, Guangzhou 510640, China

**Abstract.** Secondary organic aerosol (SOA) has great impacts on air quality, climate change and human health. The composition and physicochemical properties of SOA differ greatly because they form under different atmospheric conditions and from various precursors as well as differing oxidation. In this work, photooxidation experiments of toluene were performed under four conditions (dry, dry with $SO_2$, wet, and wet with $SO_2$) to investigate the effect of $SO_2$ under different relative humidity on the composition and optical properties of SOA at wavelengths of 375 nm and 532 nm. According to our results, the increase in humidity enhances not only light absorption, but also the scattering property of the SOA. Oligomers formed through multiphase reactions might be the reason for this phenomenon. Adding $SO_2$ slightly lowers the real part of the complex refractive index, RI(n), of toluene-derived SOA ($RI(n)_{dry, SO2} < RI(n)_{dry}$, $RI(n)_{wet, SO2} < RI(n)_{wet}$), which might be a result of the partitioning of low-oxidation state products. The imaginary part of the complex refractive index, RI(k), is enhanced under dry condition with $SO_2$ compared to that of only dry condition, which might be due to acid-catalysed aldol condensation reactions. Wet condition with $SO_2$ shows the combined effect of $SO_2$ and humidity. The extinction properties of toluene-derived SOA under wet condition with $SO_2$ increased by approximately 30% compared to that of toluene-derived SOA formed under dry condition. Our results suggest that various atmospheric conditions will affect the composition and optical proprieties of SOA, which has significant implications for evaluating the impacts of SOA on the rapid formation of regional haze, global radiative balance and climate change.

## 1. Introduction

Secondary organic aerosol (SOA) accounts for a major fraction of the atmospheric fine particulate matters (PM$_{2.5}$), and has significant impacts on air quality, climate change and human health (Seinfeld and Pandis, 2006;Jimenez et al., 2009;Hallquist et al., 2009;Huang et al., 2014). Globally, SOA could influence the radiative balance directly by scattering and absorbing solar and terrestrial radiation and indirectly by impacting cloud formation and their lifetimes (IPCC, 2013;Andreae and Gelencser, 2006;Moise et al., 2015). On the urban and regional scales, SOA contributes to the degradation of visibility and adversely influences human health (Watson, 2002;Pope et al., 2002). Light-absorbing aerosols (including black carbon (BC), mineral dust, and brown carbon (BrC)) are recognized as playing important roles in climate radiative forcing because of the strong dependence of their optical properties on the aerosol composition, the complexity of their production and the poor constraints on their contribution to radiative forcing (Peng et al., 2016;Wang et al., 2013). Quantifying the optical properties of SOA is one of the key problems in accessing anthropogenic pollution data for visibility, air quality, and climate change, as well as one of the most urgent issues in atmospheric sciences (Laskin et al., 2015;Moise et al., 2015;Andreae and Gelencser, 2006).

Aromatic compounds, accounting for 20-40% v/v of gasoline fuel, are one of the representative anthropogenic volatile organic compounds (VOCs) and play an important role in the formation of tropospheric ozone and SOA (Odum, 1997;Ng et al., 2007;Zhang et al., 2015;Odum et al., 1997). Among these aromatic hydrocarbons, toluene is thought to be one of the most important SOA precursors because of its strong emissions and SOA formation potential (Hildebrandt et al., 2009;Odum et al., 1997;Odum et al., 1996;Tao et al., 2017). The optical properties of aromatics-derived SOA have received significant attention over the last decade, most of which focused on the effect of nitrogen oxides (NO$x$) on the optical properties of aromatics SOA (Li et al., 2014;Kim and Paulson, 2013;Lin et al., 2015;Nakayama et al., 2013). In particular, Li et al. (2017) studied the effects of multiphase processes on the optical properties of m-xylene SOA (Li et al., 2017c). However, the optical properties of toluene-derived SOA under various atmospheric relevant conditions remain unclear, which limits our understanding of severe and complex air pollution.

Recent studies have shown that SOA is a dominant fraction of the total atmospheric aerosol burden, particularly in heavily polluted areas (Guo et al., 2014;Tao et al., 2017). Reported field experiments have shown that the atmospheric concentration of sulphur dioxide (SO$_2$) could be up to nearly 200 ppb in heavy haze pollution episodes in China, while the formation and growth rates of SOA and sulphate, which significantly contribute to severe haze pollution, were much faster than that during clean periods (Li et al., 2017a). According to An's study, the concentrations of NO$_2$ and nitrate were also quite high in China, especially in the North China Plain (NCP) (An et al., 2019). Previous studies have mostly focused on the enhancement of SOA yields introduced by the presence of SO$_2$ for isoprene, α-pinene and anthropogenic precursors (Santiago et al., 2012;Kleindienst et al., 2006;Liggio and Li, 2013). They also revealed that an enhancing effect of SOA yields is because of the acidic aerosol products of SO$_2$ and the formation of new particles. Surratt et al. (2010) found that reactions of isoprene epoxydiols in the presence of acidic aerosols produced from the oxidation of SO$_2$ could be a substantial source of "missing urban SOA" (Surratt et al., 2010). Jaoui et al. (2008) showed that the addition of SO$_2$ would make the colour of the SOA generated via the photooxidation of α-pinene and toluene darker and browner, and oligomers and nitrogenous organic compounds were detected in the SOA extracts (Jaoui et al., 2008). Nakayama et al. (2018) investigated the effect of SO$_2$ on the optical properties of isoprene under various NO$x$ concentrations and oxidation pathways (Nakayama et al., 2015;Nakayama et al., 2018). However, more research is needed on the role of SO$_2$ in the subsequent optical properties of SOA formed from various VOCs.

Field studies have shown that haze events were often accompanied by high relative humidity (RH) (Sun et al., 2016a;Sun et al., 2016b), and organic aerosols were mostly liquid (Shiraiwa et al., 2017;Liu et al., 2017). Multiphase reactions also play an important role in SOA formation, as shown in previous studies. For example, Jia et al. (2018) have shown that the yield of toluene SOA almost doubled at relative humidity of 85 % compared to dry conditions (Jia and Xu, 2018). Li et al. (2017) found that multiphase reactions would enhance light scattering and radiative forcing of the m-xylene SOA (Li et al., 2017c).

Understanding the impact of complicated air conditions on SOA generation, especially the effect of polluting gases (e.g., $SO_2$) and phase state, is urgently needed for forecasting, assessing, and controlling air pollution.

In this work, we investigated the effect of $SO_2$ on the optical properties and chemical composition of the SOA derived from toluene under different humidity. The results will greatly help the evaluation of the toluene-derived SOA on atmospheric visibility and climate change under complicated pollution conditions. More importantly, the data in the current study will be highly useful for the simulation of models and field observations performed under various pollution conditions.

## 2. Methods

### 2.1 Smog chamber experiments and online measurements

Experiments were performed in a 5 $m^3$ dual-reactor smog chamber, the details of which were described elsewhere (Wang et al., 2015;Li et al., 2017c;Li et al., 2017b). Briefly, the reactors were placed in a thermally isolated enclosure in which the temperature and RH could be well controlled by blowers and air conditioners. The variation range of temperature was 0.5 °C, and the limit for variability of RH was below 5%. Experiments were performed using multiple UV (ultraviolet) light sources, which resulted in a similar spectrum as that of solar radiation (Wang et al., 2015). During the experiments, the concentrations of NO, $NO_2$ and $SO_2$ were measured by corresponding gas analysers (Teledyne API T200UP, and Thermo 48i). The concentration of toluene was measured by a proton transfer reaction quadrupole mass spectrometry in $H_3O^+$ mode (PTR-QMS, Ionicon). The size distributions, number concentrations and mass concentrations of the SOA were determined by a scanning mobility particle sizer (SMPS), which consisted of an electrostatic classifier (EC, TSI 3080), a differential mobility analyser (DMA, TSI 3081) and a condensation particle counter (CPC, TSI 3776). A density of 1.4 $g·cm^{-3}$ was used for the toluene SOA according to previous studies (Ng et al., 2007;Li et al., 2014).

The extinction coefficients at 532 nm were detected with custom-built cavity ring-down spectrometers (CRDS), the details of which were given previously (Wang et al., 2012;Li et al., 2014;Li et al., 2017b). The scattering, absorption, and extinction coefficients at 375 nm were measured by a photoacoustic extinctiometer (PAX-375, Droplet Measurement Technologies).

### 2.2 Experimental conditions and off-line measurements

As shown in Table 1, four sets of experiments were conducted: D, dry condition (RH < 5%, $SO_2$ <1 ppb); DS, dry condition with $SO_2$ (RH < 5%, $SO_2$ = 30-50 ppb); W, wet condition (RH > 80%, $SO_2$ < 1 ppb); and WS, wet condition with $SO_2$ (RH > 80%, $SO_2$ = 30-50 ppb). To ensure that the results were reproducible, each experiment was performed twice. Before each experiment, the chamber was flushed at least three times using zero air (AADCO 737-15) to achieve clean conditions in which the number concentration of particles lower than 50 $cm^{-3}$ and concentrations of NO$x$ and $SO_2$ lower than 1 ppb. After preparation, a known volume of organic precursor was added to the chamber by a glass U-tube through zero air. HONO was then added into the chamber by bubbling a small flow of zero air through the immediately mixed solution of 1 mL of 1 wt.% $NaNO_2$ and 2 mL of 10 wt.% $H_2SO_4$. The by-products NO and $NO_2$ were also introduced into the chamber. The concentrations of toluene in the chamber were approximately 180 ppb, while the total NO$x$ levels were approximately 500 ppb. Parenthetically, unlike NO$x$ analysers equipped with molybdenum converters, HONO was not detected in T200UP NOx Analyser with a patented high efficiency photolytic converter. $SO_2$ was introduced into the chamber from a 29.6 ppm gas cylinder. For experiments under wet condition, zero air was passed through a fuel cell humidifier (FC-200-780-7MP, Perma Pure), and the RH was above 80% before the experiments started. It is important to highlight that dew drops did not appear in the chamber during the experiments due to accurate control of the chamber temperature (± 0.5 °C). The contents in the chamber were mixed

for approximately 10 minutes, and then the UV lamps were turned on to initiate the photoreaction. When the SOA mass concentration stopped rising, the lights were switched off to stop the reaction. All experiments were conducted at $25.0 \pm 0.5$ °C. After each experiment, the SOA was collected by PTFE filters (0.2 μm, 47 mm, MILLIPORE FELP) and then extracted in 5 mL methanol in an ultrasonic bath for 30 minutes for mass spectrometry analysis and in 10 mL for UV-Vis absorption analysis.

The methanol solutions were analysed by electrospray ionization time-of-flight mass spectrometry (ESI-TOF-MS) in positive and negative mode. A typical mass-resolving power of > 30000 was achieved at $m/z$ 200, with an absolute mass error below 3 ppm. Each sample extract was measured in triplicate with an injection volume of 10 μL. The ions detected in the filter blank were subtracted, and molecular formulas were assigned to all ions in the samples having signal-to-noise ratios greater than 10. The settings for ESI positive mode were as follows: capillary voltage, 3 kV; nebulizer gas pressure, 0.4 bar; dry gas flow rate,

3 liters per minute; dry gas temperature, 180 °C; and mass detection range, $m/z$ 100-1000. For negative mode, the capillary voltage was set to 2.5 kV, the nebulizer gas pressure was 0.3 bar, the dry gas flow rate was 4 liters per minute, the dry gas temperature was 150 °C, and the mass detection ranged from $m/z$ 200 to 1000.

The following chemicals were used without further purification: toluene (≥99.5%) [Aladdin], sulphuric acid (≥95%) [Beijing Chemical Works], sodium nitrite (98%) [Alfa Aesar], methanol (99.9%) [Fisher Chemical], and $SO_2$ ($29.6 \times 10^{-6}$ mol/mol)

[National Institute of Metrology].

**2.3 Calculation of the complex refractive indexes**

The complex refractive index (RI, RI = n + ki) is the only intrinsic optical property of particles, including the real part of complex refractive index (RI(n), standing for scattering) and the imaginary part of complex refractive index (RI(k), standing for absorption) (Bohren and Huffman, 1983;Bond and Bergstrom, 2006). The RI calculation method used in this study has

been applied and approved in previous studies (Li et al., 2014;Li et al., 2017b;Li et al., 2017c;Li et al., 2018b;Peng et al., 2018;Li et al., 2018a). Briefly, the extinction coefficients ($\alpha_{ext}$) at 532 nm measured by CRDS and the scattering, absorption, and extinction coefficients at 375 nm measured by PAX-375 were calculated on the basis of Mie theory. For monodispersed spherical particles, $\alpha_{ext}$ can be represented by:

$$\alpha_{ext} = N\sigma_{ext} = \frac{1}{4}N\pi D^2 Q_{ext} \quad (1)$$

where $\sigma_{ext}$ is the extinction cross-section, N is the number concentration of particles, D is the diameter of particles, and $Q_{ext}$ is

the extinction efficiency, which is the ratio of the extinction cross-section to the geometric area of the particles.

For polydispersed particles with log-normal size distribution and a geometric standard deviation ($\sigma_g$) smaller than 1.5, $\alpha_{ext}$ can be expressed as follows:

$$\alpha_{ext} = \frac{1}{4}S_{tot}Q_{ext} \quad (2)$$

where $S_{tot}$ is the total surface concentration. A hypothesis was made for simplifying data processing: the $Q_{ext}$ value of the polydispersed particles in the whole size distribution range was the same as the $Q_{ext}$ value of particles with the surface mean

diameter ($D_{sm}$). The $S_{tot}$ and $D_{sm}$ were given by SMPS directly, of which the uncertainties were $\pm 1\%$ and $\pm 5\%$, respectively. The value of $Q_{ext}$ of particles with a given $D_{sm}$ was subsequently calculated through the Mie program. The best-fit complex index was determined by minimizing the following reduced merit function ($\chi_r$):

$$x_r = \frac{1}{N}\sum_{i=1}^{N}(Q_{ext,measured} - Q_{ext,calculated}(n,k))_i^2 \quad (3)$$

**2.4 Calculation of RI of products**

The RI(n) of the products was calculated by the quantitative structure-property relationship, developed by Redmond and Thomson (Redmond and Thompson, 2011). In brief, the RI(n) values were estimated by polarizability (α), degree of unsaturation (μ), mass density ($\rho_m$) and molecular weight (M) via the application of equation (4).

$$RI(n)_{predicted} = 0.031717(\mu) + 0.0006087(\alpha) - 3.0227\left(\frac{\rho_m}{M}\right) + 1.38708 \qquad (4)$$

The mass density of each molecule was estimated by the E-AIM model (Extended Aerosol Inorganics Model, http://www.aim.env.uea.ac.uk/aim/density/density.php), which had high accuracy for 166 organic compounds (Girolami, 1994). It should be noted that the model should be used at 550 nm. It would cause only small, acceptable errors for its use at 532 nm. This method was employed in our previous studies as well (Li et al., 2017b;Li et al., 2018b;Li et al., 2017c;Peng et al., 2018). Similar to previous studies, negligible light absorption was found for toluene SOA at wavelength > 500 nm, as shown in Figure S3 as well, so only the real part of RI is considered at 532 nm (Li et al., 2017b;Li et al., 2018b;Li et al., 2014). For brevity, at 532 nm, "RI" represents the real part of the complex refractive index, RI(n), in the following text.

**3. Results and discussion**

**3.1 General results of the experiments**

As shown in Figure 1, particles were generated only several minutes after the lamps turned on. Although the concentrations of NO differed under different conditions, the maximum concentrations of NO$x$ were approximately the same, ~500 ppb. The consumption of the organic precursor was similar in all experiments (~100 ppb), indicating similar OH exposures and oxidation levels. As shown in Figure S1, the maximum total number concentrations of the toluene SOA were almost doubled under conditions with $SO_2$, which implied that the addition of $SO_2$ could promote new particle formation. At the end of the experiments, the $S_{tot}$ of particles were approximately $2.9 \times 10^9$, $4.3 \times 10^9$, $3.8 \times 10^9$, $6.9 \times 10^9$ nm$^2 \cdot$cm$^{-3}$ under the D, DS, W and WS conditions, respectively. The addition of $SO_2$ caused an enhancement of the total surface concentrations of the particles, probably because of the formation of new particles and/or acidic-catalysed reactions, as implied in previous studies (Chu et al., 2016;Deng et al., 2017;Hallquist et al., 2009). Moreover, the $S_{tot}$ under wet conditions was higher than that under dry conditions as well, probably due to aqueous reactions of water soluble products, as indicated previously (Jia and Xu, 2018;Kamens et al., 2011;Hinks et al., 2018).

The extinction efficiencies of the SOA under four different conditions at 375 nm and 532 nm are illustrated in Figure 2. The fitting curves match well with the experiment data points under each condition. At 532 nm, the average values of RI(n) were 1.412, 1.348, 1.504 and 1.468 under the D, DS, W and WS conditions, respectively. The complex refractive indexes at 375 nm under the D, DS, W and WS conditions were 1.45, 1.37+0.014i, 1.4566+0.022i and 1.51+0.012i, respectively. The complex refractive indexes of the SOA as a function of surface mean diameter ($D_{sm}$) under various conditions are shown in Figure S2. As the reaction proceeded, the surface mean diameter of particles increased while RI(n) decreased. This phenomenon was observed in all experiments in this study and agreed well with previous studies (Li et al., 2014;Li et al., 2017b;Peng et al., 2018).

The complex refractive indexes under dry conditions (Figure 2 (a) and Figure 3 (a)) were comparable with those reported in previous studies (Nakayama et al., 2013;Kim and Paulson, 2013;Li et al., 2014;Moise et al., 2015). For example, Kim et al. (2013) discovered that the RI(n) values of toluene-derived SOA at 532 nm were in the range of 1.35 to 1.61 (experimental conditions: 3.2 ppm toluene and 3.1 ppm NO$x$) (Kim et al., 2010;Kim and Paulson, 2013). Li et al. (2014) found RI(n) of

toluene-derived SOA was approximately 1.5 under HONO condition, and the concentration of NO$x$ was approximately 1 ppm (Li et al., 2014). The slightly lower RI(n) in the current study (1.412) might be a result of the lower NO$x$ level, as Nakayama et al. (2013) found that higher NO$x$ levels would lead to higher RI(n) values (Nakayama et al., 2013).

As shown in Figure S3, the absorption of toluene-derived SOA under the D condition at 375 nm was approximately 0.002 a.u., from which we could calculate that RI(k) under the D condition as approximately 0.001(Sun et al., 2007). PAX-375 showed no absorption of the toluene-derived SOA under dry condition, in other words, the absorption of SOA under dry condition was below the detection limit, and the calculated RI(k) was approximately 0.0009. Under this circumstance, RI(k) was set to zero (±0.001), which was lower than other studies. Table S1 lists the complex refractive indexes of toluene-derived SOA. For instance, Nakayama et al. (2013) reported that RI(k) of toluene-derived SOA was 0.05 at 355 nm and 0.002 - 0.007 at 405 nm (experimental conditions: 4.0 ppm toluene and 0.1-0.6 ppm NO$x$) (Nakayama et al., 2010;Nakayama et al., 2013), and Liu et al. (2015) suggested that the RI(k) of toluene-derived SOA was 0.01-0.03 at 320 nm and 0.002-0.02 at 405 nm (experimental conditions: 5.0 ppm toluene and 0-10 ppm NO$x$) (Liu et al., 2015). This phenomenon may have occurred because the ratios of NO to NO$_2$ during the formation of SOA were high, and light-absorbing substances, such as nitrophenols, would be hardly produced in our system in that case.

As shown in Figure 3, RI(n) and RI(k) at 532 nm and 375 nm under the four different conditions differ greatly. At 375 nm, RI(k) under the W condition was the highest, while RI(k) under the DS and WS conditions were the second highest, approximately 0.014. For RI(n), the highest value appeared in the W condition, while the lowest value showed in the DS condition. All values of RI(n) at 375 nm were larger than those at 532 nm with difference of 0.022-0.062. The real and imaginary parts of RI increased as the wavelength decreases, which was in good accordance with previous studies (Moise et al., 2015;Flores et al., 2014a;Flores et al., 2014b;Lin et al., 2015;Liu et al., 2016a;Liu et al., 2015;Nakayama et al., 2013;Nakayama et al., 2018;Nakayama et al., 2015). For example, Liu et al. (2015) found the values of RI(n) of toluene-derived secondary organic material decreased by approximately 0.1 with wavelength changed from 240 nm to 800 nm (Liu et al., 2015). In this situation, when concerning the optical properties of SOA, the wavelength should be considered as well.

To explain the RI variations of toluene-derived SOA under the different conditions, aerosols were collected and analysed by ESI-TOF-MS both in positive and negative mode. Mass spectrometry has enabled the molecular characterization of organic oligomers in SOA because of its high resolution and mass accuracy (Lin et al., 2014;Lin et al., 2012;Hinks et al., 2018). The products were inferred based on mass spectra (MS) and previous studies (Kelly et al., 2010;Hinks et al., 2018;Liu et al., 2016b;Nakayama et al., 2015). Most products were concentrated around $m/z$ 200-400 (Figures 4, S4, S5 and S6; Tables 2, S2 and S3), and some of them contained organonitrogen groups, which could be seen more clearly in negative mode. However, previous studies mainly focused on a lower mass range (< 200 Da) (Hinks et al., 2018;Forstner et al., 1997;Staudt et al., 2014;Wang et al., 2018;Birdsall et al., 2010), e.g., Birdsall et. al. (2010) detected benzaldehyde, cresol, phenol, and butenedial in the photooxidation process of toluene under both low-NO$x$ and high-NO$x$ conditions (Birdsall et al., 2010). In this work, we analysed the products around $m/z$ 200-400 and discussed the effect of SO$_2$ under different humidity. The representative identified molecular weight, molecular formula, and calculated RI(n) values of toluene-derived SOA are shown in Figure 5 and Table 2. More details are given in Table S2 and Table S3.

### 3.2 Effects of high humidity

High relative humidity causes a substantial increase in the RI(n) values for the toluene-derived SOA (Figure 2 and Figure 3). The real part of complex refractive indexes increased from 1.45 to 1.566 at 375 nm and 1.412 to 1.504 at 532 nm in absence of SO$_2$. Previous studies reported similar findings, e.g., Li et al. (2017) found that high RH would enhance the light scattering of SOA by multiphase reactions (Li et al., 2017c); Liu et al. (2016) discovered that mass absorption coefficient values at 365nm

of SOA produced by photooxidation of trimethylbenzene and toluene were enhanced when reaction RH increased from <5% to 80% (Liu et al., 2016a).

Wet conditions led to higher RI(n) values compared with those of dry conditions and induced the formation of oligomers, as shown in Figure 5 and S4. Figure S4 and previous studies (Li et al., 2017c;Liu et al., 2016a) have shown that oligomers above 500 Da appear to have high relative intensities under wet conditions, and these products possess high RI(n) values, as shown in Tables 2, S2 and S3. Multiphase reactions in the presence of liquid water led to the formation of oligomers from intermediate products such as glyoxal and methylglyoxal, resulting in large enhancements in RI(n), which was proven in previous studies (Li et al., 2017c;Liu et al., 2016a).

As proved in the study of Moltein et al. (Molteni et al., 2018), highly oxygenated molecules (HOMs) were also important oxidation products of toluene. HOMs that contained a structure of phenyl ring could not absorb light above 300 nm, however, the subsequent HOMs of nitration and ring-opening might absorb UV-Vis light. HOMs were detected in our experiments, especially under high humidity conditions. Oligomers produced from multiphase reactions, such as acid-catalysed aldol condensations, were found in mass spectra from products derived under the W condition, especially in positive mode, which could extend light absorption to longer wavelengths (Lambe et al., 2013). Lambe et al. (2013) found that the conjugated double bonds could enhance the absorption in the UV-Vis light irradiation as well (Lambe et al., 2013). This phenomenon was also proved in previous reports (Jia and Xu, 2018;Faust et al., 2017;Liu et al., 2018b).

## 3.3 Effects of SO$_2$ under dry condition

Figures 2 and 3 show that adding SO$_2$ resulted in a decrease in the RI(n) values of toluene-derived SOA from 1.45 to 1.37 at 375 nm and from 1.412 to 1.348 at 532 nm under dry condition. Compared with that under the D condition, RI(k) under the DS condition increased by 0.014. A similar phenomenon was also found in other systems, e.g., Nakayama et al. (2015) discovered that adding SO$_2$ caused the RI(n) values of isoprene-derived SOA to decrease, while the RI(k) values to increase (Nakayama et al., 2015). Nakayama et al. (2018) further found that adding SO$_2$ would cause significant light absorption at short visible and ultraviolet wavelengths of isoprene-derived SOA under OH oxidation progresses (Nakayama et al., 2018), which was quite similar with our results.

To clearly see the change of chemical compositions under the D and DS conditions, the subtraction plots of mass spectra are shown in Figure S6. As we can see in Figure S6 (a), adding SO$_2$ caused the relative intensities of products with lower molecular weight (< 200 Da) to increase. These products are mainly alcohols and esters with smaller molecular weight and RI(n) values of which are lower than 1.4. The reason for this is most likely that the addition of SO$_2$ caused large amounts of new particle formation and high particle number concentrations in the system (Chu et al., 2015;Chu et al., 2016;Deng et al., 2017;Liu et al., 2018a), which could provide a larger mass concentration and adsorb high-volatile small molecules into the particle phase (Li et al., 2018b). These small molecule products usually possess high volatility and low oxidation state, which would also reduce the oxidation state of aerosols generated by toluene. Previous studies have found similar phenomena, for example, Zhao et al. (2018) found that high SO$_2$ concentrations decreased the ratio O/C in α-pinene or limonene systems (Zhao et al., 2018), while Liu et al. (2016) discovered that the oxidation state of carbon was -0.51±0.06 for SOA formed from light-duty gasoline vehicle exhaust with SO$_2$ and -0.19±0.08 without SO$_2$ (Liu et al., 2016c), which all implied that adding SO$_2$ reduced the oxidation state of the resulting SOA. It should be noted that our off-line analytical method caused the loss of a large part of alcohols and esters with small molecular weights, which might overestimate the values of their RI(n).

Light absorption properties of SOA are related to its composition, the contribution of each product to light absorption and so on (Laskin et al., 2015;Moise et al., 2015). Ji et al. have reported that toluene oxidation proceeded dominantly via the cresol pathway and formed highly functionalized products such as polyhydroxytoluenes (Ji et al., 2017), which would undergo subsequent reactions with OH to form the precursors including α-carbonyl compounds, organic acids, and other highly

oxygenated low-volatility products. Although the peak of 3-methylcatechol, one of the major polyhydroxytoluenes of toluene oxidation products, appeared around 275 nm in UV-Vis spectrum (Ferris et al., 1971), the subsequent products like nitrocatechol and low-volatility oligomers formed by small α-carbonyl compounds might absorb light near-ultraviolet and visible. Parenthetically, the reaction rate of 3-methylcatechol with OH radicals was $7.44 \times 10^{-11}$ cm$^3$/molecule·s, and the estimation of atmospheric lifetime was 22 min approximately (Coeur-Tourneur et al., 2010). Small α-carbonyl compounds and organic acids might undergo particle phase reactions, e.g., acid-catalysed aldol condensation reactions under SO$_2$ conditions, plausibly contributing to the observed light absorption (Fu et al., 2009;Fu et al., 2008). Nakayama et al. (2018) reported the same phenomenon on isoprene SOA (Nakayama et al., 2018;Nakayama et al., 2015), while Marrero-Ortiz et al. (2019) found BrC particles were formed from small α-dicardonyls and amines (Marrero-Ortiz et al., 2019). Formation of charge transfer (CT) complexes might be another reason for light absorption enhancement, which could lead to optical transitions through a transfer of charge from a donor group, such as hydroxyls, to an acceptor group, such as a ketone or aldehyde (Phillips and Smith, 2014). For organosulfate, another kind of BrC, we did not detect them under the DS condition, which is in accordance with previous studies (Staudt et al., 2014).

### 3.4 Effects of SO$_2$ under wet condition

Under wet condition with SO$_2$, the average RI(n) values of toluene-derived SOA were 1.51 at 375 nm and 1.468 at 532 nm, which were higher than those of the DS condition and lower than those of the W condition (Figure 2 and 3). As for absorption, RI(k) under the WS condition was lower than the W condition, and similar to that under the DS condition.

Figures 4 and S5 show the results of mass spectra difference of toluene-derived SOA under the DS or W condition minuses the WS condition. Low-oxidation state organic matters and oligomers were both found in the mass spectra. Under the WS condition, relative intensities of products above 400 Da (oligomers) were higher than those under the DS condition (negative values), suggesting more types of oligomers were produced. More types of low organic matters were observed compared to those under wet condition without SO$_2$. Under these combined effects, the values of RI(n) of toluene-derived SOA of the WS condition were lower than those under the W condition and higher than those under the DS condition.

The RI(k) under the WS condition is almost equal to that under the DS condition, while lower than the values under the W condition. Products with lower oxidation state and less conjugated oligomers caused by addition of SO$_2$ might be the reason for this phenomenon, as proved in previous studies (Nakayama et al., 2015;Liu et al., 2016a). The concentrations of the donor group under the WS condition were lower than those under the DS condition, resulting in lower concentrations of CT complexes, which might be another reason for reducing the values of RI(k). Organosulfate compounds were not found under the WS condition. The combined effect of SO$_2$ and wet condition on optical properties of toluene-derived SOA are first described to the best of our knowledge, and has a significant influence on light absorption, extinction, visibility and direct radiative forcing of regional air, especially in complex polluted area. Our results provide some explanations for the observed variation, and further research is needed to quantify this synergistic effect.

### 4. Atmospheric and Climate Implication

The values of mass cross-section (MAC) were calculated for the DS, W and WS conditions at 375 nm, and the method was described in Supporting Information. The SOA under the D conditions was not calculated because no absorption was observed. The average values of MAC were 0.2749, 0.3082 and 0.2131 g/m$^2$ under the DS, W and WS conditions, respectively. These results are similar to Liu's work, which noted that wet conditions would cause higher values of MAC than those observed under dry conditions (Liu et al., 2016a). However, the MAC values in Liu's work were approximately 0.4 g/m$^2$ under wet

conditions and 0.01 $g/m^2$ at dry conditions at 380 nm, higher than our results, which is likely due to the high ratios of NO to $NO_2$ in their study.

The impacts of the atmospheric and climate were assessed by comparing the ratio of light extinction efficiency and simple forcing efficiency (SFE) under the four different conditions, and the results are shown in Figures 6 and S7. Aerosol sizes between 100 nm and 250 nm were circled because these are atmospherically relevant aerosols sizes (Zhang et al., 2015;Tao et al., 2017). As shown in Figure 6, adding $SO_2$ caused a light extinction efficiency reduction of approximately 16%-35% with an average of 25% at 532 nm, and humid conditions enhanced the light extinction efficiency by approximately 36%-64%, with an average of 50%. For the comprehensive impact made by $SO_2$ and high humidity, the light extinction efficiency increased about 16%-47% with an average of 30%. These results confirm that SOA generated under synergistic pollution conditions, conditions that contained high emissions of primary particles and gaseous precursors from multiple sources, efficient secondary matter formation, as well as adverse meteorological and climate conditions and regional transport, might have a greater impact on the visibility reduction, atmospheric photochemical reactions and secondary species formation (Dickerson et al., 1997;An et al., 2019).

We estimate the clear-sky direct radiative forcing per unit optical depth with the help of the SFE concept, as mentioned in the Supporting Information. Figure S7 shows the SFE of toluene-derived SOA under the D, DS, W and WS conditions at 375 nm and 532 nm, which could also reflect the change of direct radiative forcing (DRF). The forcing efficiency crossed over from negative (warming) to positive (cooling) values at diameter ≈ 200 nm under the DS, W and WS conditions. The lowest SFE values, approximately -35 W/g, were observed under the W condition, while at diameter approximately 200 nm, the SFE values were -10 and -20 W/g under the WS and DS conditions, respectively. All values of SFE at 532 nm were above zero, which indicates that toluene-derived SOA shows cooling effects on climate change. The SFE values under the DS condition decreased by approximately 25% compared to that under the D condition, and the SFE values under the W and WS conditions increased by approximately 50% and 30% compared to that under the D condition, respectively, which is a similar trend to that of the extinction efficiency. The SFE values in our system coincided with those of organic aerosols from wood combustion and the burning of boreal peatlands (Chen and Bond, 2010;Chakrabarty et al., 2016). Our model does not include hygroscopicity and other factors, which would increase particle size and negative forcing. In this situation, our forcing is much lower than those in global climate models (Schulz et al., 2006). The combined effects of $SO_2$ and humidity should be considered in the modified climate model.

## 5. Conclusion

The effect of $SO_2$ under different humidity on the optical properties of SOA photooxidized by toluene was investigated in this study. The results show that for the experimental system with RH greater than 80%, as expected, the increase in humidity greatly enhanced the real part of RI, from 1.412 to 1.504 at 532nm and from 1.45 to 1.566 at 375 nm, the imaginary part of RI was enhanced as well, which is probably because of the oligomers formation from multiphase reactions. Adding $SO_2$ can reduce the RI(n) values of toluene-derived SOA at 375 nm and 532 nm, whether under low or high humidity. The RI values of toluene-derived SOA produced under dry conditions with $SO_2$ are 1.37+0.014i at 375 nm and 1.348 at 532 nm, while the RI(n) values under dry and $SO_2$-free conditions are 1.412 at 532 nm and 1.45 at 375 nm. The reason for this phenomenon might be that adding $SO_2$ caused large amounts of new particle formation and high particle mass concentrations in the system, which could have absorbed high-volatile molecules into the particle phase. High-volatile molecules produce lower oxidation state and lower RI(n) values, resulting in the decrease of RI(n) for toluene-derived SOA. The increase in RI(k) is probably related to acid-catalysed reactions on acidic particles. For the experimental system under high humidity condition with sulfur dioxide, the RI(n) was higher than SOA derived from dry condition without $SO_2$. The RI(k) under this condition are lower

than those under wet condition without $SO_2$ because fewer oligomers formed. The extinction properties under the WS condition are approximately 30% higher than under the D condition. The results here highlighted that the combined effect of $SO_2$ and high humidity could greatly enhance the refractive index, light scattering, and direct radiative forcing of toluene-derived SOA and, potentially, others. These results will improve our understanding of SOA optical properties, especially under complex atmospheric conditions.

**Author contributions.**  WW, MG and WZ conceived and led the studies. WZ, JL and CP performed chamber simulation and data analysis. KL, LZ, BS, YC and ML discussed the results and commented on the manuscript. WZ prepared the manuscript with contributions from all co-authors.

**Acknowledgements**

This project was supported by The National Key Research and Development Program of China (2016YFC0202704), National research program for key issues in air pollution control (DQGG-0103)
and the National Natural Science Foundation of China (91544227, 91844301, 41822703).

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

**Table 1. Initial conditions and retrieved average RI values of the SOA formed under different sets of experiments.**

| Experiments No. | HC (ppb) | NO (ppb) | $NO_X$ (ppb) | $SO_2$ (ppb) | RH (%) | RI average (375 nm) | RI(n) average (532 nm) |
|---|---|---|---|---|---|---|---|
| D1 | 182 | 247 | 351 | <1 | <5 | 1.45 | 1.412 |
| D2 | 196 | 283 | 425 | <1 | <5 | 1.458 | 1.414 |
| DS1 | 193 | 278 | 425 | 30 | <5 | 1.37+0.014i | 1.348 |
| DS2 | 160 | 407 | 647 | 50 | <5 | 1.366+0.014i | 1.344 |
| W1 | 170 | 241 | 361 | <1 | >80 | 1.566+0.022i | 1.504 |
| W2 | 172 | 235 | 351 | <1 | >80 | 1.562+0.018i | 1.508 |
| WS1 | 165 | 229 | 261 | 35 | >80 | 1.51+0.012i | 1.468 |
| WS2 | 185 | 257 | 399 | 37 | >80 | 1.528+0.01i | 1.474 |

30

**Table 2. Representative identified mass spectral peaks, molecular weights, formulas and calculated RI(n) values of the main products of toluene-derived SOA.**

| Type | M + H | M + Na | M − H | Molecular Formula | Calculated RI |
|---|---|---|---|---|---|
| D |  | 142.94 |  | $C_4H_8O_5$ | 1.4070 |
| D | 137.06 |  |  | $C_4H_9NO_2$ | 1.4073 |
| D | 162.04 |  |  | $C_7H_{15}NO_3$ | 1.4072 |
| D | 218.92 |  |  | $C_{11}H_{22}O_4$ | 1.4173 |
| D | 226.14 |  |  | $C_7H_{15}NO_7$ | 1.4124 |
| D |  |  | 241.23 | $C_7H_{14}O_9$ | 1.4173 |
| D |  |  | 253.23 | $C_9H_{18}O_8$ | 1.4166 |
| D | 261.23 |  |  | $C_{14}H_{28}O_4$ | 1.4265 |
| D |  |  | 269.26 | $C_9H_{18}O_9$ | 1.4175 |
| D |  |  | 273.04 | $C_{15}H_{30}O_4$ | 1.4264 |
| D | 352.24 |  |  | $C_{16}H_{33}NO_7$ | 1.4303 |
| D | 360.31 |  |  | $C_{12}H_{25}NO_{11}$ | 1.4268 |
| D |  |  | 367.37 | $C_{16}H_{32}O_9$ | 1.4318 |
| D | 419.31 | 441.30 |  | $C_{15}H_{30}O_{13}$ | 1.4330 |
| DS |  | 113.10 |  | $C_4H_{10}O_6$ | 1.3595 |
| DS | 107.97 | 130.15 |  | $C_3H_9NO_3$ | 1.3727 |
| DS | 198.11 |  |  | $C_6H_{15}NO_6$ | 1.3776 |
| DS | 212.15 |  |  | $C_7H_{16}O_7$ | 1.3800 |
| DS |  |  | 233.14 | $C_{12}H_{26}O_4$ | 1.3918 |
| DS |  |  | 254.24 | $C_9H_{21}NO_7$ | 1.3911 |
| DS |  |  | 291.14 | $C_{15}H_{32}O_5$ | 1.3950 |
| DS | 305.25 |  |  | $C_9H_{20}O_{11}$ | 1.3893 |
| DS |  |  | 337.19 | $C_{16}H_{35}O_7$ | 1.3997 |
| DS |  |  | 346.02 | $C_{11}H_{25}NO_{11}$ | 1.3972 |
| DS |  |  | 353.18 | $C_{16}H_{34}O_8$ | 1.4004 |
| DS |  |  | 381.21 | $C_{19}H_{32}O_8$ | 1.4032 |
| DS |  |  | 397.20 | $C_{18}H_{38}O_9$ | 1.4032 |
| DS | 455.31 |  |  | $C_{22}H_{46}O_9$ | 1.4085 |
| DS | 475.38 |  |  | $C_{20}H_{42}O_{12}$ | 1.4066 |
| DS |  |  | 514.36 | $C_{23}H_{49}NO_{11}$ | 1.4122 |
| W |  |  | 223.02 | $C_{12}H_{16}O_4$ | 1.5459 |
| W | 318.24 |  |  | $C_{14}H_{27}NO_7$ | 1.5227 |
| W |  |  | 323.23 | $C_{11}H_{16}O_{11}$ | 1.5230 |
| W | 346.33 |  |  | $C_{16}H_{27}NO_7$ | 1.5258 |
| W |  |  | 422.18 | $C_{17}H_{29}NO_{11}$ | 1.5313 |
| W |  |  | 459.28 | $C_{23}H_{40}O_9$ | 1.5364 |
| W |  |  | 467.04 | $C_{19}H_{32}O_{13}$ | 1.5334 |

| | | | | |
|---|---|---|---|---|
| **W** | 509.25 | | $C_{23}H_{40}O_{12}$ | 1.5374 |
| **W** | | 523.22 | $C_{23}H_{40}O_{13}$ | 1.5380 |
| **W** | | 541.05 | $C_{22}H_{38}O_{15}$ | 1.5366 |
| **W** | 559.50 | | $C_{22}H_{38}O_{16}$ | 1.5382 |
| **W** | 583.51 | | $C_{26}H_{46}O_{14}$ | 1.5423 |
| **W** | 615.15 | | $C_{26}H_{46}O_{16}$ | 1.5432 |
| **W** | 641.60 | | $C_{29}H_{52}O_{15}$ | 1.5465 |
| **W** | 642.60 | | $C_{28}H_{51}NO_{15}$ | 1.5461 |
| **WS** | 203.05 | | $C_9H_{14}O_5$ | 1.4793 |
| **WS** | | 217.04 | $C_9H_{14}O_6$ | 1.4762 |
| **WS** | | 271.09 | $C_{14}H_{24}O_5$ | 1.4904 |
| **WS** | | 279.04 | $C_{10}H_{16}O_9$ | 1.4875 |
| **WS** | | 301.04 | $C_{15}H_{26}O_6$ | 1.4918 |
| **WS** | 315.16 | | $C_{17}H_{30}O_5$ | 1.4950 |
| **WS** | | 317.06 | $C_{15}H_{26}O_7$ | 1.4926 |
| **WS** | | 331.06 | $C_{16}H_{28}O_7$ | 1.4941 |
| **WS** | | 341.09 | $C_{11}H_{18}O_{12}$ | 1.4878 |
| **WS** | 365.11 | | $C_{16}H_{28}O_9$ | 1.4974 |
| **WS** | | 379.06 | $C_{16}H_{28}O_{10}$ | 1.4954 |
| **WS** | 409.29 | | $C_{18}H_{32}O_{10}$ | 1.4974 |
| **WS** | | 409.08 | $C_{17}H_{30}O_{11}$ | 1.4974 |
| **WS** | | 415.02 | $C_{14}H_{24}O_{14}$ | 1.4945 |
| **WS** | | 429.03 | $C_{15}H_{26}O_{14}$ | 1.4945 |
| **WS** | | 461.04 | $C_{23}H_{42}O_9$ | 1.5036 |
| **WS** | | 483.04 | $C_{20}H_{36}O_{13}$ | 1.5032 |
| **WS** | 522.60 | | $C_{25}H_{47}NO_{10}$ | 1.5081 |
| **WS** | 550.70 | | $C_{27}H_{51}NO_{10}$ | 1.5109 |
| **WS** | | 580.98 | $C_{27}H_{50}O_{13}$ | 1.5126 |
| **WS** | | 597.21 | $C_{27}H_{50}O_{14}$ | 1.5125 |

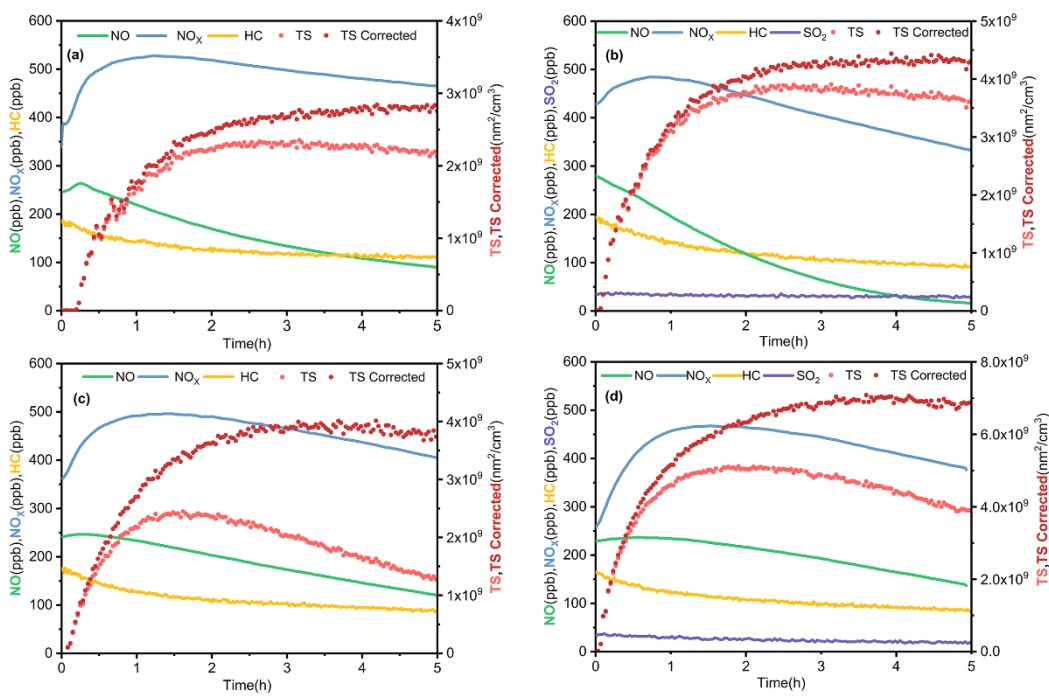

**Figure 1.** Reaction profiles of photooxidation of toluene under four different conditions: (a) D1 (dry condition), (b) DS1 (dry condition with $SO_2$), (c) W1 (wet condition) and (d) WS1 (wet condition with $SO_2$); see Table 1 for details. The concentrations of gas species are shown on the left axis, while the concentrations of particles are shown on the right axis.

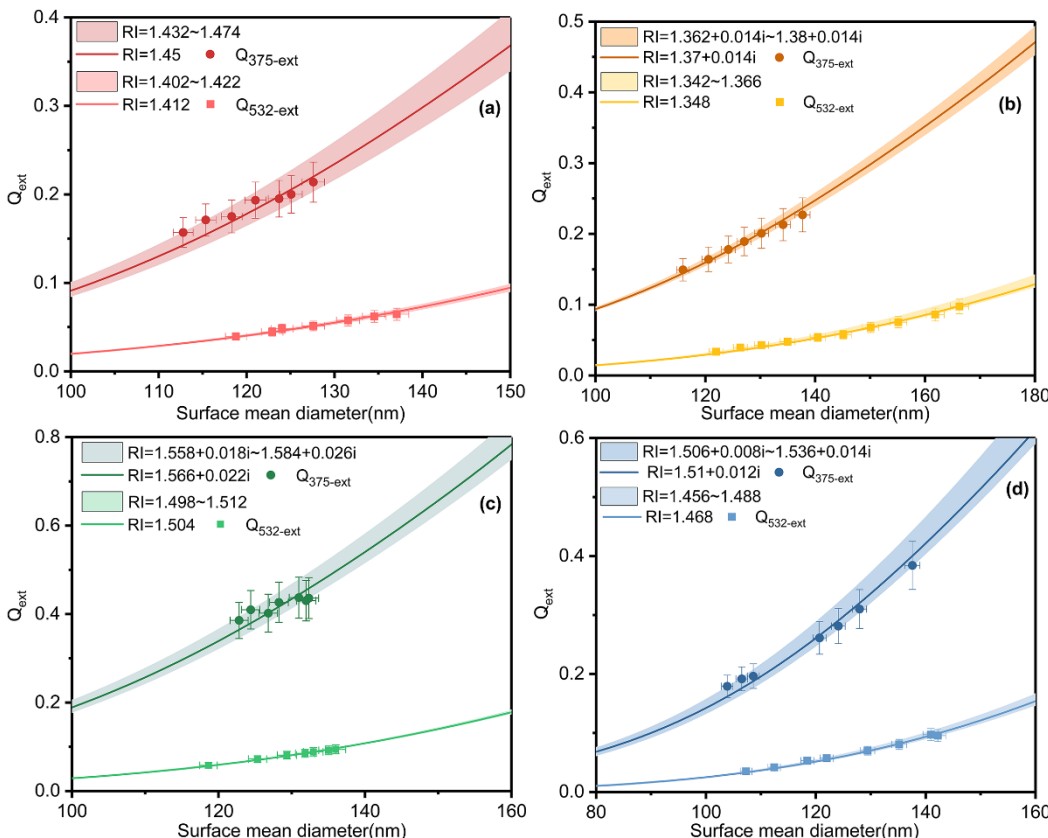

**Figure 2. Dependence of the extinction efficiencies of the SOA on the surface mean diameter under four different conditions of (a) D1, (b) DS1, (c) W1 and (d) WS1. The lines are results retrieved from the average RI, while the shadows are ranges of the retrieved RI.**

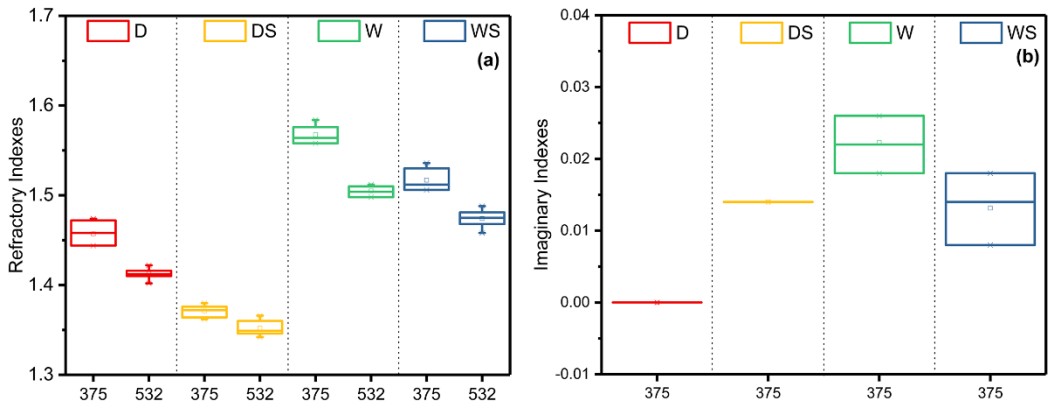

**Figure 3. The difference of the values of (a) RI(n) and (b) RI(k) for toluene-derived SOA under the four different conditions at 375 nm and 532 nm. The values of RI(k) are zero at 532 nm and not shown in this figure.**

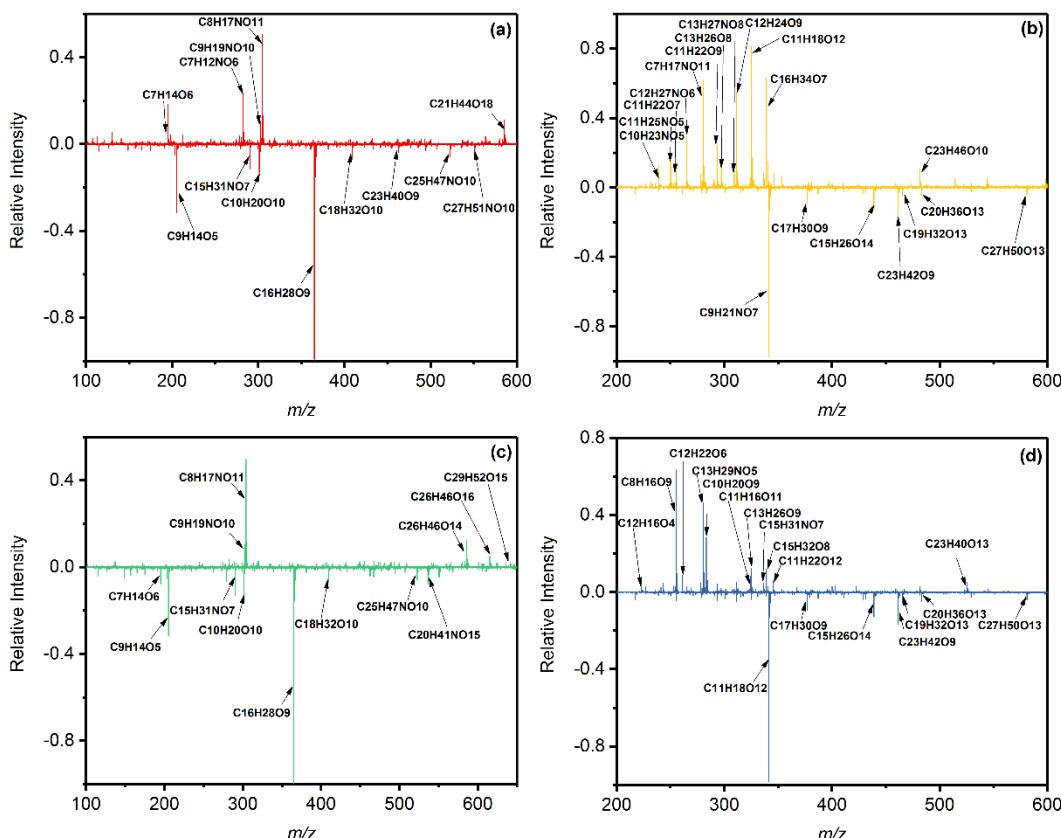

**Figure 4. Results of mass spectra difference of toluene SOA under the DS condition minuses the WS condition in (a) positive and (b) negative mode, and of the W condition minuses the WS condition in (c) positive and (d) negative mode. Y axis is the subtraction of relative intensity (indicated by the peak intensity relative to the strongest peak intensity) between condition DS or W and WS.**

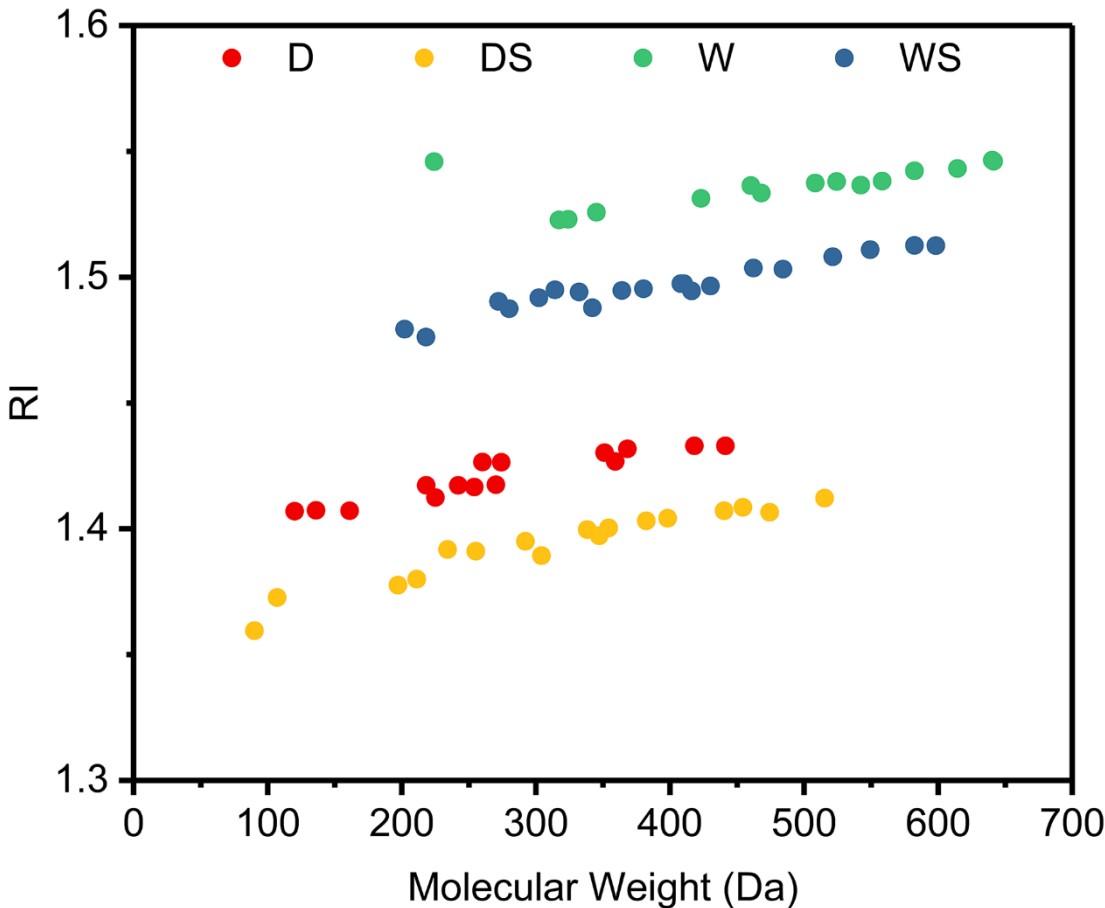

**Figure 5. The representative molecular weights and calculated RI values under the four conditions.**

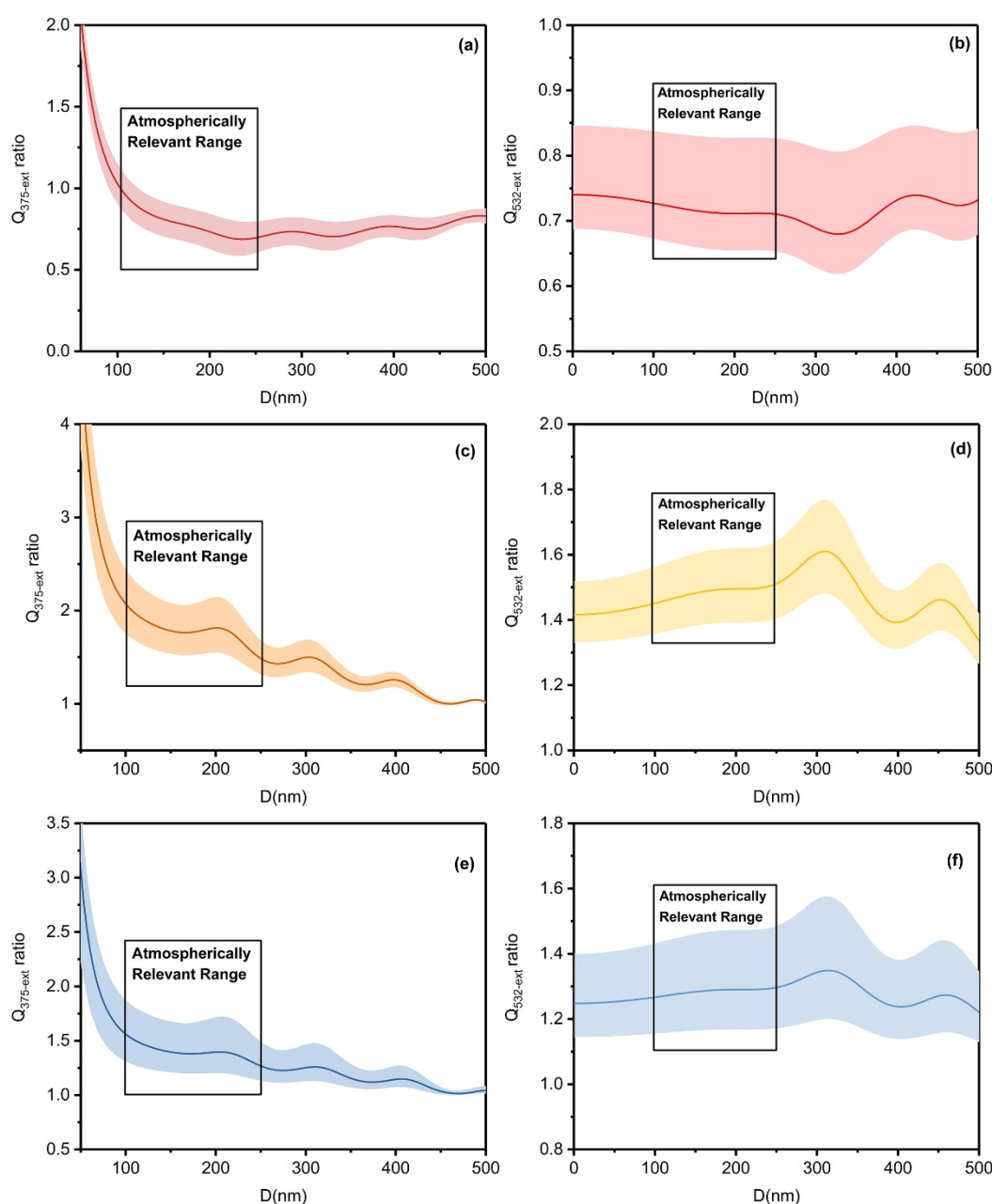

**Figure 6. The ratio of light extinction efficiency of the toluene-derived aerosol of the DS condition to that of the D condition at (a) 375 nm and (b) 532 nm, of the W condition to the D condition at (c) 375nm and (d) at 532nm, of the WS condition to the D condition at (e) 375 nm and (f) 532 nm. The lines are the average values, the shaded areas are uncertainties and the box areas represent atmospherically relevant ranges.**

