# Peer review of "Effects of SO2 on optical properties of secondary organic aerosol generated from photooxidation of toluene under different relative humidity"

_Atmospheric Chemistry and Physics, 2019_

## Referee Comment (RC1) · Anonymous Referee #1 · 19 Aug 2019

General comments:

The authors reported the effects of relative humidity and sulfur dioxide level on the optical properties of secondary organic aerosol from the toluene/NOx photooxidation. Toluene is a representative anthropogenic aromatic hydrocarbon. Secondary organic aerosol from anthropogenic aromatic hydrocarbons comprises a portion of ambient organic aerosol particles at a global scale, and may affect the earth's climate. The optical properties of toluene secondary organic aerosol were already reported by several groups. However, a dataset reported in this manuscript is valuable because it is poorly understood on the effects of relative humidity and sulfur dioxide level on the optical

properties of toluene secondary organic aerosol. In the current form, discussion will be insufficient on (1) discrepancy between present and previous results of the visible absorption of SOA formed under dry neutral conditions and (2) the effects of humidity and acidity on charge transfer complexes. Revisions are necessary for the publication.

Major comments:

(1) The authors reported that secondary organic aerosol formed under dry neutral conditions had little absorption at 375 nm. In contrast, several previous studies reported that toluene secondary organic aerosol formed under dry neutral conditions had visible absorption. Nakayama et al. (2010; 2013) reported that the imaginary refractive index (k) was 0.05 at 355 nm and 0.002 – 0.007 at 405 nm, where aerosol was formed at RH <1%. Zhong and Jang (2011) reported that k was 0.02 at 350 nm in the absence of seed particles, where aerosol was formed at RH = 42 – 43%. Liu et al. (2015) reported that k was 0.01 – 0.03 at 320 nm and 0.002 – 0.02 at 405 nm, where aerosol was formed at RH = 13%. The author should compare present results with these previous results to characterize present experimental conditions. The reviewer assumes that nitrophenols, light-absorbing substances formed during classical toluene/NOx experiments, will barely be produced in their toluene/HONO/NOx irradiation system due to high NO to NO2 ratios during secondary aerosol formation.

(2) The authors suggest that visible absorption of secondary organic aerosol, formed in the toluene/NOx/SO2 irradiation system, was attributed to charge transfer complexes. They refer results of ambient aerosol collected in the winter season for the assignment of light-absorbing substances present in laboratory toluene secondary aerosol samples. The authors suggest that charge transfer complexes are formed between small alcohol and carbonyl molecules. However, reactions of alcohols with carbonyls in the condensed phase are enhanced under acidic conditions to result in the formation of hemiacetals. Furthermore, the reactions of carbonyls with aerosol water will lead to the formation of hydrated carbonyls, which cannot form charge transfer complexes with alcohols. It is widely accepted that the formation of hemiacetals and hydrated carbonyls

occur during secondary organic aerosol formation (Jang et al., 2002). The formation of hemiacetals and hydrated carbonyls will suppress the formation of charge transfer complexes under humid and acidic conditions; this is inconsistent with present trends of k, which increased with increasing humidity and acidity. The authors should discuss the effects of humidity and acidity on chemical reactions between alcohols and carbonyls in the condensed phase and might need to tone down the identifications of charge transfer complexes as light-absorption substances.

(3) In line 33, page 4, the authors define RI as complex refractive index, n + ik. However, RI is often used as the real part of refractive index (n) in the latter part of the text. Please distinguish between RI and n through text. For example, all RIs used in eq. 4 and lines 8, 17, and 19 of page 6 must indicate the real part of refractive index. Also, "RRI" in line 29, page 6 must be the real part of refractive index. There will also be multi-definition RI symbols at other places.

(4) Line 35, page 6 and other places. The authors use the terminologies, "macro-molecules" and "polymers," for high molecular weight products form the oxidation of toluene. Generally, macromolecules and polymers are defined to be high molecular compounds in excess of 1,000 atoms (Staudinger and Fritschi, 1922), indicating that the molecular weight of macromolecules is higher than several thousands. On the other hand, the molecular weight is reported to be less than 1,000 for high molecular weight products formed from the toluene oxidation (Sato et al., 2007; Molteni et al., 2018). From this point of view, high molecular weight compounds detected in secondary organic aerosol particles are generally referred as oligomers. Please consider using "oligomers" instead of "organic macromolecules", "macromolecular polymers", or "macromolecular oligomers" for high-molecular weight products formed from the photooxidation of toluene.

(5) There are many minor grammatical errors through text. For example, "secondary organic aerosol (SOA) account for. . ." in line 2, page 2 should be "secondary organic aerosol (SOA) accounts for. . .". In line 26, page 3, the description, "SMPS, which

was consisted of. . .,"should be "SMPS, which consisted of . . .". The reviewer does not point out all errors. The reviewer recommends the authors to use commercial English correction.

Specific comments:

(6) Line 12, page 4., What is "a total NOx level" when a mixture of HONO, NO, and NO2 is measured by a NOx monitor? Please explain it explicitly. Is it a total level of pure NOx (= NO + NO2)?

(7) Line 14, page 4. The relative humidity was set to more than 80% during experiments under humid conditions. The reviewer believes that dew drops may appear on chamber wall under such high relative humidity conditions, and water soluble small organic compounds will be dissolved into these dew drops. Loss of small organic compounds might affect results of the optical properties of secondary organic aerosol. It should be described whether dew drops appeared during experiments or not.

(8) Line 18, page 5. What is AIM? Please explain this abbreviation.

(9) The first paragraph of section 3.1 and Fig. 1. Why did the NO concentration decreased much faster under a dry acidic condition than for other conditions?

(10) Lines 30 – 32, page 6. The authors describe "the optical properties of secondary organic aerosol should be concerned differently at different wavelength." The meaning of this sentence is unclear. Please rewrite it.

(11) Line 38, page 6. Generally, "200 – 400 m/z" is written as "m/z 200 – 400." According to IUPAC, m/z should be italic.

(12) Lines 15 – 16, page 7. The authors describe "oligomers above 500 Da appear high relative intensities." Please show mass spectra obtained for experiments under dry and humid neutral conditions.

(13) Lines 1 – 2, page 8. How did the authors identify butanehexol, cyclohexanepentol,

and methylpentitol? Experimental data shown for chemical identifications in this study are only mass spectra observed by direct sample infusion into the mass spectrometer. They could only suggest chemical formulae form observed mass spectra. The results of concentrations of alcohols are used as evidences for the formation of charge transfer complexes in this manuscript. The identifications of these alcohol products should be explained in detail.

(14) Lines 2 – 5, page 8. The gas-particle partitioning is determined by the aerosol mass concentration rather than the surface concentration (Pankow, 1994). The rate of reactive uptake due to surface reactions may be determined by the aerosol surface concentration.

(15) The second paragraph of section 3.4 and Figure 4. The authors describe only that Figures 4a and 4b are difference mass spectra between DS and WS. Please explain whether these figures show results of "DS – WS" or "WS – DS"? Similar explanations should be added for Figures 4c, 4d, S2a, and S2b. In addition, chemical formulae shown in these figures are too small. These should be enlarged.

(16) Lines 32 – 33, page 8. Which results show that the oligomer concentrations observed for WS experiments are higher than for DS? Both figures 4a and 4b show difference signals are very small for products with m/z > 400, suggesting that there was no big difference in the oligomer concentrations between WS and DS.

(17) Line 23, page 9. What is "complex pollution"? It should be explained specifically.

(18) Lines 1 – 2, page 10. The authors describe "a full climate model would be necessary to determine the actual forcing caused by this (these?) effects caused by SO2 and humidity as well." The meaning of this sentence is unclear. Please rewrite it.

(19) Lines 6 – 7, page 10. If the authors want to emphasize that the increase of the real part of refractive index, please discuss using real pert data only.

(20) Line 15, page 10. The authors specify "dehydration reactions" in the conclusions,

but they did not discuss whether oligomerization process contains dehydration or not in the main text.

References:

Jang, M., Czoschke, N. M., Lee, S., & Kamens, R. M. (2002). Heterogeneous atmospheric aerosol production by acid-catalyzed particle-phase reactions. Science, 298(5594), 814-817.

Liu, P., Abdelmalki, N., Hung, H. M., Wang, Y., Brune, W. H., & Martin, S. T. (2015). Ultraviolet and visible complex refractive indices of secondary organic material produced by photooxidation of the aromatic compounds toluene and m-xylene. Atmospheric Chemistry and Physics, 15(3), 1435-1446.

Molteni, U., Bianchi, F., Klein, F., Haddad, I. E., Frege, C., Rossi, M. J., ... & Baltensperger, U. (2018). Formation of highly oxygenated organic molecules from aromatic compounds. Atmospheric Chemistry and Physics, 18(3), 1909-1921.

Nakayama, T., Matsumi, Y., Sato, K., Imamura, T., Yamazaki, A., & Uchiyama, A. (2010). Laboratory studies on optical properties of secondary organic aerosols generated during the photooxidation of toluene and the ozonolysis of $\alpha$‐pinene. Journal of Geophysical Research: Atmospheres, 115(D24), doi:10.1029/2010JD014387.

Nakayama, T., Sato, K., Matsumi, Y., Imamura, T., Yamazaki, A., & Uchiyama, A. (2013). Wavelength and NO x dependent complex refractive index of SOAs generated from the photooxidation of toluene. Atmospheric Chemistry and Physics, 13(2), 531-545.

Pankow, J. F. (1994). An absorption model of gas/particle partitioning of organic compounds in the atmosphere. Atmospheric Environment, 28(2), 185-188.

Sato, K., Hatakeyama, S., & Imamura, T. (2007). Secondary organic aerosol formation during the photooxidation of toluene: NO x dependence of chemical composition. The Journal of Physical Chemistry A, 111(39), 9796-9808.

Staudinger, H., & Fritschi, J. (1922). Über Isopren und Kautschuk. 5. Mitteilung. Über die Hydrierung des Kautschuks und über seine Konstitution. Helvetica Chimica Acta, 5(5), 785-806.

Zhong, M., & Jang, M. (2011). Light absorption coefficient measurement of SOA using a UV–Visible spectrometer connected with an integrating sphere. Atmospheric environment, 45(25), 4263-4271.

---

## Referee Comment (RC2) · Anonymous Referee #2 · 17 Sep 2019

In this article the authors investigated the effects of SO2 and RH on the optical properties of secondary organic aerosol (SOA) produced by toluene photooxidation using smog chamber experiments. Photooxidation of toluene was performed in a 5 m3 dual-reactor smog chamber under the condition of dry (D), dry with SO2 (DS), wet (W), or wet with SO2 (WS). Optical properties of SOA were measured by a photoacoustic extinctiometer (PAX) and a cavity ring-down spectrometer (CRDS) at the wavelength of 375 nm as well as 532 nm. Particle composition was measured by an electrospray ionization time-of-flight mass spectrometry. The authors found that RH enhanced light absorption and scattering of SOA and indicated that it was due to the formation of highly conjugated oligomers formed through multiphase reactions. They also found

that adding SO2 slightly lowered the real part of complex refractive index (n) but increased the imaginary part (k) for dry condition and explained by the partitioning of low oxidation state products and the formation of charge transfer complexes with SO2 appeared. The authors concluded that their results have significant implication for evaluating the impacts of SOA on the regional haze, global radiative balance, and climate change. Overall, the topic is suitable for the readership of Atmos. Chem. Phys. I recommend publication of this work, provided that the following issues have been adequately addressed. The authors explained the differences in the optical properties with adding SO2 by larger amounts of new particles formation and higher particle surface concentrations. Since particle concentration and size distribution were measured in the experiments, direct evidence of NPF needed to be provided to support this argument. Also, an increase of particle surface concentration increases the rate of uptake but could not help to stabilize the volatile compounds in the particle phase. There needs some discussion on the chemical mechanism leading to their measured optical properties of secondary organic aerosol. For example, it has been recently proposed that the toluene oxidation proceeds dominantly via the cresol pathway (Ji et al., Reassessing the atmospheric oxidation mechanism of toluene, Proc. Natl. Acad. Sci. USA 114, 8169–8174, 2017). How do the different mechanisms of toluene oxidation impact their optical measurements and conclusions. Also, toluene oxidation is known as a major source for small alpha-dicarbonyls (glyoxal and methylglyoxal), which can be a source for brown carbon (Marrero-Ortiz et al., Formation and Optical Properties of Brown Carbon from Small $\alpha$-Dicarbonyls and Amines, Environ. Sci. Technol. 53(1), 117-126, 2019). Such an aspect needs to be addressed in this paper. Under different conditions, the peak of the absorption spectrum can shift due to the different functionalities, which could also be a reason for different optical properties at certain wavelength. Since the PAX and CRDS only measure the optical properties at 375 nm and 532 nm, it would help if the full spectrum is provided. In the introduction, the role of light-absorption aerosols on radiative forcing and pollution development needs to be discussed. In particularly, several key references are missing here (An et al., Severe haze in Northern

China: A synergy of anthropogenic emissions and atmospheric processes, Proc. Natl. Acad. Sci. USA 116, 8657–8666, 2019; Peng et al., Markedly enhanced absorption and direct radiative forcing of black carbon under polluted urban environments, Proc. Natl. Acad. Sci. USA 113, 4266–4271, 2016; Wang et al., Light absorbing aerosols and their atmospheric impacts, Atmos. Environ. 81, 713-715, 2013). Those studies have discussed the atmospheric impacts of light-absorbing. It would also be necessary that discussions are provided to compare the results from this work to those of literature. Minor points

The font size of chemical formulas in Fig. 4 is too small to be read clearly.

P7 Line 32, "that" is repeated.
* * *

---

## Author Comment (AC1) · 30 Oct 2019

**Responses to Referee #1:**

We thank the reviewer for thoughtful and constructive suggestions that help us improve our manuscript. According to the comments, we have added the related statements in the Results and Discussion section. In the following, please find our responses to the comments one by one and the corresponding revisions made to the manuscript (highlighted in yellow color). The original comments are shown in italics, and our point-by-point responses are listed below. We believe that the current form of the manuscript is much improved, and we hope the new version is now suitable for publication in ACP.

*General comments:*

*The authors reported the effects of relative humidity and sulfur dioxide level on the optical properties of secondary organic aerosol from the toluene/NOx photooxidation. Toluene is a representative anthropogenic aromatic hydrocarbon. Secondary organic aerosol from anthropogenic aromatic hydrocarbons comprises a portion of ambient organic aerosol particles at a global scale, and may affect the earth's climate. The optical properties of toluene secondary organic aerosol were already reported by several groups. However, a dataset reported in this manuscript is valuable because it is poorly understood on the effects of relative humidity and sulfur dioxide level on the optical properties of toluene secondary organic aerosol. In the current form, discussion will be insufficient on (1) discrepancy between present and previous results of the visible absorption of SOA formed under dry neutral conditions and (2) the effects of humidity and acidity on charge transfer complexes. Revisions are necessary for the publication.*

Response: Thanks for the positive and constructive comments, we have carefully revised the manuscript and details are listed below.

(1) In order to compare the results of scattering and absorption of toluene SOA, especially under dry conditions, we have added a table about the reaction conditions, mass concentrations and refractive indexes of toluene SOA, as well as the UV-Vis absorption spectrum of toluene SOA (Table S1 and Figure S3).

**Table S1.** Reaction conditions, instruments, mass concentrations and refractive indexes of SOA derived from toluene.

| Reaction conditions | | | | SOA properties | | | Wavelength ($\lambda$/nm) | Reference |
|---|---|---|---|---|---|---|---|---|
| Toluene (ppm) | NO (ppm) | NO$x$ (ppm) | Oxidation (ppm) | Mass ($\mu$g/m$^3$) | Refractive index | | | |
| | | | | | RI(n) | RI(k) | | |
| ~10 | \ | \ | O$_3$, ~10 | \ | 1.49-1.50 | \ | 532 | Redmond and Thompson et al., 2011 |
| 0.063-0.168 | \ | 0.03-0.12 | O$_3$, ~10 | \ | SSA, 0.95 | | 355 | Ma et al., 2012 |
| 1 | \ | \ | O$_3$, ~20 | 592 | 1.45 | 0.024 | 405 | Feng et al., 2019 |
| 1.7-3.7 | 0.6-3.1 | 0.6-3.1 | propene | 74-680 | 1.35-1.61 | \ | 532 | Kim and Paulson et al., 2010 Kim et al., 2011 |
| 4.0 | \ | 0.5 | CH$_3$ONO, ~0.01 | 127-216 | 1.632 | 0.047 | 355 | Nayakama et al., 2010 |
| | | | | | 1.483 | 0.007 | 532 | |
| 4.0 | 0.1-0.6 | 0.1-0.6 | CH$_3$ONO, ~0.01 | 30-160 | 1.449-1.567 | 0.0018-0.0072 | 405 | Nayakama et al., 2013 |
| | | | | | 1.431-1.498 | 0.0000-0.0010 | 532 | |
| | | | | | 1.389-1.452 | 0.0000 | 781 | |
| 0.2 | \ | \ | H$_2$O$_2$, 5 | \ | 1.45 | \ | 532 | Li et al.,2014 |
| 0.34 | \ | \ | H$_2$O$_2$, ~16; 30%RH | | \ | 0.0005 | 365 | Liu et al., 2016 |
| 0.34 | 1.8 | 1.8 | H$_2$O$_2$, ~16; 30%RH | | \ | 0.019 | 365 | |
| 0.4 | 1.8 | 1.8 | H$_2$O$_2$, ~16; <5%RH | \ | | | | |
| 0.3 | 1.8 | 1.8 | H$_2$O$_2$, ~16; 50%RH | | \ | | \ | |
| 0.34 | 1.8 | 1.8 | H$_2$O$_2$, ~16; 80%RH | | | | | |
| 0.2 | \ | 0.07 | None seed | 47 | | 0.02 | 350 | Zhong and Jang, 2011 |
| | | | AS | 38-44 | \ | 0.06 | | |
| | | | Acid | 31-35 | | 0.045 | | |
| 4.0 | 0.04 | 0.4 | \ | \ | 1.464 | 0.000 | 532 | Li et al.,2014 |
| 0.2 | 0.2 | 0.9 | \ | | 1.518 | 0.000 | | |
| 5 | 0-10 | 0-10 | \ | 770-2000 | 1.567-1.597 | 0.011-0.033 | 320 | Liu et al., 2015 |
| | | | | | 1.546-1.571 | 0.002-0.015 | 405 | |
| 0.18 | 0.24 | 0.35 | RH<5% | | 1.45 | 0.000 | 375 | this work |
| | | | | | 1.412 | 0.000 | 532 | |
| 0.19 | 0.28 | 0.43 | SO$_2$=30ppb, <5%RH | | 1.37 | 0.014 | 375 | |
| | | | | \ | 1.348 | 0.000 | 532 | |
| 0.17 | 0.24 | 0.36 | >80%RH | | 1.566 | 0.022 | 375 | |
| | | | | | 1.504 | 0.000 | 532 | |
| 0.18 | 0.23 | 0.26 | SO$_2$=40 ppb, >80%RH | | 1.51 | 0.12 | 375 | |
| | | | | | 1.468 | 0.000 | 532 | |

[Figure]

**Figure S3.** UV-Vis absorption spectrum of toluene SOA under the four different conditions.

According to Figure S3, the absorption under condition D was approximately 0.002 a.u., and the imaginary part of refractive indexes can be calculated by equation (1) (Sun et al., 2007):

$$k = \frac{ln(10)}{4\pi} \frac{\rho\lambda}{cL} A(\lambda) \tag{1}$$

where $\rho$ is the material density (g cm$^{-3}$), as for toluene SOA, $\rho$ is set to 1.4 g cm$^{-3}$ (Ng et al., 2007); c is the concentration of solution (g cm$^{-3}$), L is optical path length (cm), $\lambda$ is wavelength (cm$^{-1}$). The calculated RI(k) under the D condition was 0.001, while the calculated RI(k) under the DS, W and WS conditions were 0.009, 0.019 and 0.008 respectively. These results have good coincidences with the results of PAX-375. And the detection limit of PAX-375 was 0.5 Mm$^{-1}$, from which we could calculate the RI(k) under the D condition is 0.0009 according to Mie theory. Under this circumstance, RI(k) is set to zero ($\pm$0.001), which is lower than other studies, as shown in Table S1. This phenomenon may have occurred because the ratios of NO to $NO_2$ during the formation

of SOA were high, and light-absorbing substances, such as nitrophenols, would be hardly produced in our system in that case.

We also have added discussion "As shown in Figure S3, the absorption of toluene-derived SOA under the D condition at 375 nm was approximately 0.002 a.u., from which we could calculate that RI(k) under the D condition as approximately 0.001(Sun et al., 2007). PAX-375 showed no absorption of the toluene-derived SOA under dry condition, in other words, the absorption of SOA under dry condition was below the detection limit, and the calculated RI(k) was approximately 0.0009. Under this circumstance, RI(k) was set to zero (±0.001), which was lower than other studies. Table S1 lists the complex refractive indexes of toluene-derived SOA. For instance, Nakayama et al. (2013) reported that RI(k) of toluene-derived SOA was 0.05 at 355 nm and 0.002 - 0.007 at 405 nm (experimental conditions: 4.0 ppm toluene and 0.1-0.6 ppm $NO_x$) (Nakayama et al., 2010; Nakayama et al., 2013), and Liu et al. (2015) suggested that the RI(k) of toluene-derived SOA was 0.01-0.03 at 320 nm and 0.002-0.02 at 405 nm (experimental conditions: 5.0 ppm toluene and 0-10 ppm $NO_x$) (Liu et al., 2015). This phenomenon may have occurred because the ratios of NO to $NO_2$ during the formation of SOA were high, and light-absorbing substances, such as nitrophenols, would be hardly produced in our system in that case." (Line 3-15, paragraph 4 in Section 3.1, page 7) in the new version.

(2) We have reassessed the effects of humidity and acidity on charge transfer complexes, and rewrote the discussion to "Light absorption properties of SOA are related to its composition, the contribution of each product to light absorption and so on (Laskin et al., 2015; Moise et al., 2015). Small α-dicardonyls compounds such as glyoxal and methylglyoxal are important intermediate products of toluene that undergo polymerization to produce low-volatility oligomers (Ji et al., 2017; Fu et al., 2009; Fu et al., 2008). These products might undergo particle phase reactions, e.g., acid-catalysed aldol condensation reactions under $SO_2$ conditions, plausibly contributing to the observed light absorption. Nakayama et al. (2018) reported the same phenomenon on isoprene SOA (Nakayama et al., 2018; Nakayama et al., 2015), while Marrero-Ortiz et al. (2019) found BrC particles were formed from small α-dicardonyls and amines

(Marrero-Ortiz et al., 2019). Formation of charge transfer (CT) complexes might be another reason for light absorption enhancement, which could lead to optical transitions through a transfer of charge from a donor group, such as hydroxyls, to an acceptor group, such as a ketone or aldehyde (Phillips and Smith, 2014). For organosulfate, another kind of BrC, we did not detect them under the DS condition, which is in accordance with previous studies (Staudt et al., 2014)." (Line 11-24, paragraph 3 in Section 3.3, page 9). And we have revised the abstract part to "……The imaginary part of the complex refractive index, RI(k), is enhanced under dry condition with $SO_2$ compared to that of only dry condition, which might be due to acid-catalysed aldol condensation reactions.……" (Line 11-14, page 1) In conclusion part, we have rewritten to "……The increase in RI(k) is probably related to acid-catalysed reactions on acidic particles.……" (Line 23-24, page 11).

Reference:

Ng, N. L., Kroll, J. H., Chan, A. W. H., Chhabra, P. S., Flagan, R. C., and Seinfeld, J. H.: Secondary organic aerosol formation from m-xylene, toluene, and benzene, Atmospheric Chemistry and Physics, 7, 3909-3922, 2007.

Sun, H., Biedermann, L., and Bond, T. C.: Color of brown carbon: A model for ultraviolet and visible light absorption by organic carbon aerosol, Geophysical Research Letters, 34, 10.1029/2007gl029797, 2007.

*Major comments:*

*(1) The authors reported that secondary organic aerosol formed under dry neutral conditions had little absorption at 375 nm. In contrast, several previous studies reported that toluene secondary organic aerosol formed under dry neutral conditions had visible absorption. Nakayama et al. (2010; 2013) reported that the imaginary refractive index (k) was 0.05 at 355 nm and 0.002 − 0.007 at 405 nm, where aerosol was formed at RH <1%. Zhong and Jang (2011) reported that k was 0.02 at 350 nm in the absence of seed particles, where aerosol was formed at RH = 42 − 43%. Liu et al. (2015) reported that k was 0.01 − 0.03 at 320 nm and 0.002 − 0.02 at 405 nm, where aerosol was formed at RH = 13%. The author should compare present results with these previous results to characterize present experimental conditions. The reviewer*

*assumes that nitrophenols, light-absorbing substances formed during classical toluene/NOx experiments, will barely be produced in their toluene/HONO/NOx irradiation system due to high NO to NO$_2$ ratios during secondary aerosol formation.*

Response: Thank you very much for pointing out these important issues. PAX-375 showed no absorption of SOA under dry condition, and the detection limit of absorption of PAX-375 was approximately 0.5Mm$^{-1}$. Under this circumstance, we calculated that RI(k) was below 0.0009. We also conducted the UV-Vis absorption spectrum measurements of toluene SOA under the four different conditions to further validation (Figure S3). As shown in Figure S3, the absorption of toluene under the D condition was approximately 0.002 a.u., the calculated RI(k) of toluene SOA was about 0.001. These results were lower than other studies, as shown in Table S1, a list of complex refractive indexes of toluene SOA. This phenomenon might attribute to high NO to NO$_2$ ratios during formation of SOA. In this case, light-absorbing substances, such as nitrophenols, would be barely produced in our system.

So we added "As shown in Figure S3, the absorption of toluene-derived SOA under the D condition at 375 nm was approximately 0.002 a.u., from which we could calculate that RI(k) under the D condition as approximately 0.001(Sun et al., 2007). PAX-375 showed no absorption of the toluene-derived SOA under dry condition, in other words, the absorption of SOA under dry condition was below the detection limit, and the calculated RI(k) was approximately 0.0009. Under this circumstance, RI(k) was set to zero (±0.001), which was lower than other studies. Table S1 lists the complex refractive indexes of toluene-derived SOA. For instance, Nakayama et al. (2013) reported that RI(k) of toluene-derived SOA was 0.05 at 355 nm and 0.002 - 0.007 at 405 nm (experimental conditions: 4.0 ppm toluene and 0.1-0.6 ppm NO$x$) (Nakayama et al., 2010; Nakayama et al., 2013), and Liu et al. (2015) suggested that the RI(k) of toluene-derived SOA was 0.01-0.03 at 320 nm and 0.002-0.02 at 405 nm (experimental conditions: 5.0 ppm toluene and 0-10 ppm NO$x$) (Liu et al., 2015). This phenomenon may have occurred because the ratios of NO to NO$_2$ during the formation of SOA were

high, and light-absorbing substances, such as nitrophenols, would be hardly produced in our system in that case." (Line 3-15, paragraph 4 in Section 3.1, page 7).

*(2) The authors suggest that visible absorption of secondary organic aerosol, formed in the toluene/NOx/SO2 irradiation system, was attributed to charge transfer complexes. They refer results of ambient aerosol collected in the winter season for the assignment of light-absorbing substances present in laboratory toluene secondary aerosol samples. The authors suggest that charge transfer complexes are formed between small alcohol and carbonyl molecules. However, reactions of alcohols with carbonyls in the condensed phase are enhanced under acidic conditions to result in the formation of hemiacetals. Furthermore, the reactions of carbonyls with aerosol water will lead to the formation of hydrated carbonyls, which cannot form charge transfer complexes with alcohols. It is widely accepted that the formation of hemiacetals and hydrated carbonyls occur during secondary organic aerosol formation (Jang et al., 2002). The formation of hemiacetals and hydrated carbonyls will suppress the formation of charge transfer complexes under humid and acidic conditions; this is inconsistent with present trends of k, which increased with increasing humidity and acidity. The authors should discuss the effects of humidity and acidity on chemical reactions between alcohols and carbonyls in the condensed phase and might need to tone down the identifications of charge transfer complexes as light-absorption substances.*

Response: Thanks for the suggestion. We have read the reference and compared with the reaction conditions, and we changed the paragraph to "Light absorption properties of SOA are related to its composition, the contribution of each product to light absorption and so on (Laskin et al., 2015; Moise et al., 2015). Small $\alpha$-dicardonyls compounds such as glyoxal and methylglyoxal are important intermediate products of toluene that undergo polymerization to produce low-volatility oligomers (Ji et al., 2017; Fu et al., 2009; Fu et al., 2008). These products might undergo particle phase reactions, e.g., acid-catalysed aldol condensation reactions under $SO_2$ conditions, plausibly

contributing to the observed light absorption. Nakayama et al. (2018) reported the same phenomenon on isoprene SOA (Nakayama et al., 2018; Nakayama et al., 2015), while Marrero-Ortiz et al. (2019) found BrC particles were formed from small α-dicardonyls and amines (Marrero-Ortiz et al., 2019). Formation of charge transfer (CT) complexes might be another reason for light absorption enhancement, which could lead to optical transitions through a transfer of charge from a donor group, such as hydroxyls, to an acceptor group, such as a ketone or aldehyde (Phillips and Smith, 2014). For organosulfate, another kind of BrC, we did not detect them under the DS condition, which is in accordance with previous studies (Staudt et al., 2014)." (Line 11-24, paragraph 3 in Section 3.3, page 9). And we have revised the abstract part to "……The imaginary part of the complex refractive index, RI(k), is enhanced under dry condition with $SO_2$ compared to that of only dry condition, which might be due to acid-catalysed aldol condensation reactions.……" (Line 11-14, page 1) In conclusion part, we have rewritten to "……The increase in RI(k) is probably related to acid-catalysed reactions on acidic particles.……" (Line 23-24, page 11).

*(3) In line 33, page 4, the authors define RI as complex refractive index, n + ik. However, RI is often used as the real part of refractive index (n) in the latter part of the text. Please distinguish between RI and n through text. For example, all RIs used in eq. 4 and lines 8, 17, and 19 of page 6 must indicate the real part of refractive index. Also, "RRI" in line 29, page 6 must be the real part of refractive index. There will also be multi-definition RI symbols at other places.*

Response: Thanks for the suggestions. The complex refractive index, RI, is expressed as a complex number: m = n + ki, where the real part of the complex index, RI(n), represents the scattering and the imaginary part of the complex index, RI(k), represents absorption. As shown in Figure S3, there is negligible absorption at wavelengths > 500 nm, which indicated RI(k) could be set to zero at wavelengths > 500 nm. In this case, RI equals to the real part of RI, RI(n).

To prevent confusion, we have changed RI at 532 nm to RI(n) in the new version, for instance, in equation (4) and lines 23, 33 and 35, page 6, as mentioned above, and line 19 and 24, page 7 as well. We added "Same as previous studies, negligible light absorption was found for toluene SOA at wavelength > 500 nm, so only the real part of RI is considered at 532 nm (Li et al., 2017b; Li et al., 2018b; Li et al., 2014). For brevity, at 532nm, "RI" represents the real part of refractive index in the following text." (Line 1-4, page 6). And we unified the abbreviation for real part of RI for the full text, for instance, we rephrased "RRI" to RI(n) in line 26, page 6.

*(4) Line 35, page 6 and other places. The authors use the terminologies, "macromolecules" and "polymers," for high molecular weight products form the oxidation of toluene. Generally, macromolecules and polymers are defined to be high molecular compounds in excess of 1,000 atoms (Staudinger and Fritschi, 1922), indicating that the molecular weight of macromolecules is higher than several thousands. On the other hand, the molecular weight is reported to be less than 1,000 for high molecular weight products formed from the toluene oxidation (Sato et al., 2007; Molteni et al., 2018). From this point of view, high molecular weight compounds detected in secondary organic aerosol particles are generally referred as oligomers. Please consider using "oligomers" instead of "organic macromolecules", "macromolecular polymers", or "macromolecular oligomers" for high-molecular weight products formed from the photooxidation of toluene.*

Response: Thanks for the suggestions. We have read the reference and normalize the terminologies. We replaced the words "organic macromolecules", "macromolecular polymers" and "macromolecular oligomers" to "oligomers" through the article. For example, we have rephrased macromolecules in line 28, page 7, macromolecular polymers in line 11, page 8 and macromolecular oligomers in line 13, page 8 to oligomers.

*(5) There are many minor grammatical errors through text. For example, "secondary organic aerosol (SOA) account for. . ." in line 2, page 2 should be "secondary organic aerosol (SOA) accounts for. . .". In line 26, page 3, the description, "SMPS, which was consisted of. . .,"should be "SMPS, which consisted of . . .". The reviewer does not point out all errors. The reviewer recommends the authors to use commercial English correction.*

Response: We appreciate the reviewer a lot for pointing out the writing problems. We have used commercial English to help us for proper English language, grammar, punctuation, spelling, and overall style. Minor errors have been corrected, for example,

1. "Secondary organic aerosol (SOA) account for. . ." has been corrected to "Secondary organic aerosol (SOA) accounts for. . ." (Line 2, page 2).

2. "SMPS, which was consisted of. . ." has been corrected to "SMPS, which consisted of an electrostatic classifier (EC, TSI 3080), a differential mobility analyser (DMA, TSI 3081) and a condensation particle counter (CPC, TSI 3776)." (Line 34-35, page 3).

3. The singular and plural error "at wavelength of 375 nm and 532 nm" has been corrected to "at wavelengths of 375 nm and 532 nm" (Line 6, page 1).

4. "Adding $SO_2$ slightly lower the real part of complex refractive index (n) of SOA……" has been corrected to "Adding $SO_2$ slightly lowers the real part of complex refractive index (n) of SOA……" (Line 8-9, page 1).

5. "……optical proprieties of SOA, which have significant implications for……" has been corrected to "……optical proprieties of SOA, which has significant implications for……" (Line 16-17, page 1).

*Specific comments:*

*(6) Line 12, page 4., What is "a total NOx level" when a mixture of HONO, NO, and NO2 is measured by a NOx monitor? Please explain it explicitly. Is it a total level of pure NOx (= NO + NO2)?*

Response: As described in Section 2.2, HONO was produced by adding 1mL 1 wt. % NaNO$_2$ into 2mL 10 wt. % H$_2$SO$_4$, with by-products NO and NO$_2$. All the gases were introduced to the chamber and measured by NO$x$ Analyser. The concentrations of NO, NO$_2$ in our experiments were given by NO$x$ Analyser (Teledyne API T200UP) directly. NO$x$ in our manuscript was treated as the sum of NO and NO$_2$, all given by the NO$x$ Analyser. T200UP NO$x$ Analyser combined with a patented high efficiency photolytic converter, which ensured NO$_2$ to be selectively converted to NO with negligible interference from other gases. In other words, the gas flow passed through a conversion chamber and it was exposed to a blue light from two high powered ultraviolet light emitting diodes (LEDs) characterized by narrow emission bands centered at 395 nm. The emission band of the LEDs included NO$_2$ absorption band with negligible interferences expected from other gases such as HONO or NO$_3$ radicals. Consequently, unlike NO$x$ analysers equipped with molybdenum converters, HONO was not detected in T200UP NO$x$ Analyser.

*(7) Line 14, page 4. The relative humidity was set to more than 80% during experiments under humid conditions. The reviewer believes that dew drops may appear on chamber wall under such high relative humidity conditions, and water soluble small organic compounds will be dissolved into these dew drops. Loss of small organic compounds might affect results of the optical properties of secondary organic aerosol. It should be described whether dew drops appeared during experiments or not.*

Response: Thanks for the reminding. Dew drops were not appeared in the chamber for air in the chamber was well mixed by blowers and temperature was well controlled in our chamber, as previous studies reported. For example, we assumed that the relative humidity in our experiments was 85% and the temperature was 25.0 ℃. If the temperature changed from 24.5 ℃ to 25.5 ℃, the relative humidity would change from 82.5% to 87.5%. Under this circumstance, dew drops were not shown during the experiments. We added "It is important to highlight that dew drops did not appear in

the chamber during the experiments due to accurate control of the chamber temperature ($\pm 0.5$ ℃)." (Line 19-20, page 4).

*(8) Line 18, page 5. What is AIM? Please explain this abbreviation.*

Response: Thanks for the reminding. For more precise description, we have changed "AIM model" to "E-AIM model (Extended Aerosol Inorganics Model, http://www.aim.env.uea.ac.uk/aim/density/density.php)" (Line 26-27, page 5).

*(9) The first paragraph of section 3.1 and Fig. 1. Why did the NO concentration decreased much faster under a dry acidic condition than for other conditions?*

Response: As the reviewer pointing out, NO did consume more under dry acidic condition, which is very interesting. The reason for this phenomenon might be higher concentrations of toluene and $NO_2$ in this experiment compared to those in other experiments, which might convert to higher concentrations of $O_3$ during the experiments. Under this circumstance, NO might consume much faster.

*(10) Lines 30 – 32, page 6. The authors describe "the optical properties of secondary organic aerosol should be concerned differently at different wavelength." The meaning of this sentence is unclear. Please rewrite it.*

Response: Thanks for the suggestions. We rewrote the sentence to "when concerning the optical properties of SOA, the wavelength should be considered as well." (Line 25-26, page 7).

*(11) Line 38, page 6. Generally, "200 – 400 m/z" is written as "m/z 200 – 400." According to IUPAC, m/z should be italic.*

Response: Thanks for the suggestion, and all "m/z" have changed to "*m/z*" in the new version, in line 28, page 4, line 32 and 38, page 7, and Figure 4, for instance.

*(12) Lines 15 – 16, page 7. The authors describe "oligomers above 500 Da appear high relative intensities." Please show mass spectra obtained for experiments under dry and humid neutral conditions.*

Response: Thanks for the suggestions. We have added Figure S4, results of mass spectra difference of toluene SOA under the D condition minus the W condition in positive mode, oligomers above 500 Da were multiplied 5 times and showed in Figure S4b. We also added, "Figure S4 and previous studies (Li et al., 2017c; Liu et al., 2016a) have shown that ……"(Line 12-15, page 8).

[Figure]

**Figure S4.** Results of mass spectra difference of toluene SOA under the D condition minus the W condition in (a) positive mode and (b) larger version of figure S4(a), oligomers above 500 Da were multiplied 5 times. The Y axis is the subtraction of relative intensity (indicated by the peak intensity relative to the strongest peak intensity) between condition D and W.

*(13) Lines 1 – 2, page 8. How did the authors identify butanehexol, cyclohexanepentol, and methylpentitol? Experimental data shown for chemical identifications in this study are only mass spectra observed by direct sample infusion into the mass spectrometer. They could only suggest chemical formulae form observed mass spectra. The results of concentrations of alcohols are used as evidences for the formation of charge transfer*

*complexes in this manuscript. The identifications of these alcohol products should be explained in detail.*

Response: Thanks for the reminding. Butanehexol, cyclohexanepentol, and methylpentitol were identified as important products in previous studies, we assumed SOA products from our experiments to be same products for they share same chemical formula. The error has been corrected in the new version, we changed the sentence to "These products are mainly alcohols and esters with smaller molecular weight and RI(n) values of which are lower than 1.4." (Line 35-36, page 8).

*(14) Lines 2 – 5, page 8. The gas-particle partitioning is determined by the aerosol mass concentration rather than the surface concentration (Pankow, 1994). The rate of reactive uptake due to surface reactions may be determined by the aerosol surface concentration.*

Response: Thanks for the suggestions. We have revised the sentence to "The reason for this is most likely that the addition of $SO_2$ caused large amounts of new particle formation and high particle number concentrations in the system (Chu et al., 2015; Chu et al., 2016; Deng et al., 2017; Liu et al., 2018a), which could provide a larger mass concentration and adsorb high-volatile small molecules into the particle phase (Li et al., 2018b)." (Line 36-37, page 8 and Line 1-3, page 9).

*(15) The second paragraph of section 3.4 and Figure 4. The authors describe only that Figures 4a and 4b are difference mass spectra between DS and WS. Please explain whether these figures show results of "DS – WS" or "WS – DS"? Similar explanations should be added for Figures 4c, 4d, S2a, and S2b. In addition, chemical formulae shown in these figures are too small. These should be enlarged.*

Response: Thanks for the suggestions. These figures showed the subtraction plots of mass spectra of toluene SOA under the DS condition minus the WS condition and the

W condition minus the WS condition. The captions of Figures 4, S4, S5 and S6 have changed to "Results of mass spectra difference of toluene SOA under the A condition minus the B condition in positive and negative mode." And we changed the sentence to "Figures 4 and S5 show the results of mass spectra difference of toluene-derived SOA under the DS or W condition minus the WS condition." (Line 30-31, page 9). The sizes of chemical formulae in the figures have been enlarged from 12 to 16 in new figures.

*(16) Lines 32 – 33, page 8. Which results show that the oligomer concentrations observed for WS experiments are higher than for DS? Both figures 4a and 4b show difference signals are very small for products with m/z > 400, suggesting that there was no big difference in the oligomer concentrations between WS and DS.*

Response: Thanks for the suggestions. The signals in mass spectra hadn't been corrected for lack of standards, and the relative intensities might not show the real concentrations of products. Under these circumstances, we compared the relative intensities of different products between two mass spectra and identified different ion peaks. In order to show the results clearly, signals above 400 Da in mass spectra were amplified by a factor of 5 in Figure S5. From the figures, we could see that the relative intensities of products above 400 Da (oligomers) were higher under condition WS (negative values), suggesting the oligomer concentrations observed for WS experiments were higher than for DS. We added "Figures 4 and S5 showed the results of mass spectra difference of toluene SOA under condition DS or W minus condition WS." (Line 20, page 9). And more experiments would be done in the future.

[Figure]

**Figure S5.** Larger version of results of *m/z* above 400 mass spectra difference of toluene SOA under the DS condition minus the WS condition in (a) positive mode and (b) negative mode (oligomers above 400 Da were multiplied 5 times). The Y axis is the subtraction of relative intensity (indicated by the peak intensity relative to the strongest peak intensity) between condition DS and WS.

*(17) Line 23, page 9. What is "complex pollution"? It should be explained specifically.*

Response: Thanks for the reminding. To express clearly, we have changed "complex pollution conditions" to "synergistic pollution conditions", which represented the conditions that contained high emissions of primary particles and gaseous precursors from multiple sources, efficient secondary matter formation, as well as adverse meteorological and climate conditions and regional transport. We changed to "These results confirm that SOA generated under synergistic pollution conditions, conditions that contained high emissions of primary particles and gaseous precursors from multiple sources, efficient secondary matter formation, as well as adverse meteorological and climate conditions and regional transport, might have a greater impact on the visibility reduction……" (Line 27-29, page 10).

*(18) Lines 1 – 2, page 10. The authors describe "a full climate model would be necessary to determine the actual forcing caused by this (these?) effects caused by SO2 and humidity as well." The meaning of this sentence is unclear. Please rewrite it.*

Response: Thanks for the suggestion. We have rewritten the sentence to "The combined effects of $SO_2$ and humidity should be considered in the modified climate model." (Line 9-10, page 11).

*(19) Lines 6 – 7, page 10. If the authors want to emphasize that the increase of the real part of refractive index, please discuss using real pert data only.*

Response: Thanks for the suggestion. In order to avoid misunderstanding, we have rewritten this sentence to "……the increase in humidity greatly enhanced the real part of RI, from 1.412 to 1.504 at 532nm and from 1.45 to 1.566 at 375 nm, the imaginary part of RI was enhanced as well, which ……" (Line 14-15, page 11).

*(20) Line 15, page 10. The authors specify "dehydration reactions" in the conclusions, but they did not discuss whether oligomerization process contains dehydration or not in the main text.*

Response: Thanks for the suggestion. We have deleted "dehydration reactions" in the conclusion part to avoid confusion.

---

## Author Comment (AC2) · 30 Oct 2019

**Responses to Referee #2:**

We thank the reviewer for thoughtful and constructive suggestions that help us improve our manuscript. According to the comments, we have added the related statements in the Results and Discussion section. In the following, please find our responses to the comments one by one and the corresponding revisions made to the manuscript (highlighted in yellow color). The original comments are shown in italics, and our point-by-point responses are listed below. We believe that the current form of the manuscript is much improved, and we hope the new version is now suitable for publication in ACP.

*General comments:*

*In this article the authors investigated the effects of SO2 and RH on the optical properties of secondary organic aerosol (SOA) produced by toluene photooxidation using smog chamber experiments. Photooxidation of toluene was performed in a 5 m3 dualreactor smog chamber under the condition of dry (D), dry with SO2 (DS), wet (W), or wet with SO2 (WS). Optical properties of SOA were measured by a photoacoustic extinctiometer (PAX) and a cavity ring-down spectrometer (CRDS) at the wavelength of 375 nm as well as 532 nm. Particle composition was measured by an electrospray ionization time-of-flight mass spectrometry. The authors found that RH enhanced light absorption and scattering of SOA and indicated that it was due to the formation of highly conjugated oligomers formed through multiphase reactions. They also found that adding SO2 slightly lowered the real part of complex refractive index (n) but increased the imaginary part (k) for dry condition and explained by the partitioning of low oxidation state products and the formation of charge transfer complexes with SO2 appeared. The authors concluded that their results have significant implication for evaluating the impacts of SOA on the regional haze, global radiative balance, and climate change. Overall, the topic is suitable for the readership of Atmos. Chem. Phys. I recommend publication of this work, provided that the following issues have been adequately addressed.*

Response: We thank the reviewer for these positive and constructive comments.

*The authors explained the differences in the optical properties with adding $SO_2$ by larger amounts of new particles formation and higher particle surface concentrations. Since particle concentration and size distribution were measured in the experiments, direct evidence of NPF needed to be provided to support this argument. Also, an increase of particle surface concentration increases the rate of uptake but could not help to stabilize the volatile compounds in the particle phase. There needs some discussion on the chemical mechanism leading to their measured optical properties of secondary organic aerosol. For example, it has been recently proposed that the toluene oxidation proceeds dominantly via the cresol pathway (Ji et al., Reassessing the atmospheric oxidation mechanism of toluene, Proc. Natl. Acad. Sci. USA 114, 8169–8174, 2017). How do the different mechanisms of toluene oxidation impact their optical measurements and conclusions. Also, toluene oxidation is known as a major source for small alpha-dicarbonyls (glyoxal and methylglyoxal), which can be a source for brown carbon (Marrero-Ortiz et al., Formation and Optical Properties of Brown Carbon from Small - Dicarbonyls and Amines, Environ. Sci. Technol. 53(1), 117-126, 2019). Such an aspect needs to be addressed in this paper.*

Response: Thanks for the suggestions. As the previous studies reported, the addition of $SO_2$ would cause large amounts of new particle formation. We observed the same phenomenon and added a comparison diagram about the maximum of total number concentration of SOA (Figure S1). In the discussion part, we added "As shown in Figure S1, the maximum of total number concentrations of toluene SOA was almost doubled under conditions with $SO_2$, which implied that the addition of $SO_2$ could promote new particle formation." (Line 10-13, page 6).

[Figure]

**Figure S1.** The maximum of total number concentrations of SOA derived from toluene under the D, DS, W and WS conditions.

As for the mechanism, we have changed Paragraph 3 in Section 3.3 to "Light absorption properties of SOA are related to its composition, the contribution of each product to light absorption and so on (Laskin et al., 2015; Moise et al., 2015). Small α-dicardonyls compounds such as glyoxal and methylglyoxal are important intermediate products of toluene that undergo polymerization to produce low-volatility oligomers (Ji et al., 2017; Fu et al., 2009; Fu et al., 2008). These products might undergo particle phase reactions, e.g., acid-catalysed aldol condensation reactions under $SO_2$ conditions, plausibly contributing to the observed light absorption. Nakayama et al. (2018) reported the same phenomenon on isoprene SOA (Nakayama et al., 2018; Nakayama et al., 2015), while Marrero-Ortiz et al. (2019) found BrC particles were formed from small α-dicardonyls and amines (Marrero-Ortiz et al., 2019). Formation of charge transfer (CT) complexes might be another reason for light absorption enhancement, which could lead to optical transitions through a transfer of charge from a donor group, such as hydroxyls, to an acceptor group, such as a ketone or aldehyde (Phillips and Smith, 2014). For

organosulfate, another kind of BrC, we did not detect them under the DS condition, which is in accordance with previous studies (Staudt et al., 2014)." (Line 11-24, paragraph 3 in Section 3.3, page 9). And we have revised the abstract part to "……The imaginary part of the complex refractive index, RI(k), is enhanced under dry condition with $SO_2$ compared to that of only dry condition, which might be due to acid-catalysed aldol condensation reactions.……" (Line 11-14, page 1) In conclusion part, we have rewritten to "……The increase in RI(k) is probably related to acid-catalysed reactions on acidic particles.……" (Line 23-24, page 11).

*Under different conditions, the peak of the absorption spectrum can shift due to the different functionalities, which could also be a reason for different optical properties at certain wavelength. Since the PAX and CRDS only measure the optical properties at 375 nm and 532 nm, it would help if the full spectrum is provided.*

Response: The full UV-Vis absorption spectrum of toluene under four conditions has been conducted and provided in Figure S3. As shown in the figure, the absorption of toluene SOA decreased with increasing wavelength (in the range of 300–600 nm, typical range of sunlight).

[Figure]

**Figure S3.** UV-Vis absorption spectrum of toluene SOA under four different conditions.

*In the introduction, the role of light-absorption aerosols on radiative forcing and pollution development needs to be discussed. In particularly, several key references are missing here (An et al., Severe haze in Northern China: A synergy of anthropogenic emissions and atmospheric processes, Proc. Natl. Acad. Sci. USA 116, 8657–8666, 2019; Peng et al., Markedly enhanced absorption and direct radiative forcing of black carbon under polluted urban environments, Proc. Natl. Acad. Sci. USA 113, 4266–4271, 2016; Wang et al., Light absorbing aerosols and their atmospheric impacts, Atmos. Environ. 81, 713-715, 2013). Those studies have discussed the atmospheric impacts of light-absorbing. It would also be necessary that discussions are provided to compare the results from this work to those of literature.*

Response: Thanks for the suggestions. We have added "Light-absorbing aerosols (including black carbon (BC), mineral dust, and brown carbon (BrC)) are recognized as playing important roles in climate radiative forcing because of the strong dependence

of their optical properties on the aerosol composition, the complexity of their production and the poor constraints on their contribution to radiative forcing (Peng et al., 2016; Wang et al., 2013)." (Line 8-12, page 2). We also added "According to An's study, the concentrations of NO2 and nitrate were also quite high in China, especially in the North China Plain (NCP) (An et al., 2019)." (Line 30-33, page 2).

*Minor points*

*The font size of chemical formulas in Fig. 4 is too small to be read clearly.*

Response: Thanks for the suggestions. The sizes of chemical formulae have been enlarged from 12 to 16 in Figures 4, S4 and S6.

*P7 Line 32, "that" is repeated.*

Response: The mistake has been corrected.

---

## Author Response (AR2)

Dear Prof. Surratt,

Thank you very much for your support and suggestions for our manuscript. Also, we thank the reviewer again for careful reading and thoughtful suggestions that help us improve our manuscript. According to the comments, we have added the related statements in the Results and Discussion section. In the following, please find our responses to the comments one by one and the corresponding revisions made to the manuscript (highlighted in yellow color). The original comments are shown in italics, and our point-by-point responses are listed below. We believe the new version is now suitable for publication in ACP.

*The referee believes that the revised manuscript becomes much better than before and the current form will satisfy almost requirements from referees. However the following minor revisions will still be necessary for the publication:*

*(1) Referee #2 commented that it has been recently proposed that the toluene oxidation proceeds dominantly via the cresol pathway (Ji et al., 2017). The authors cited this suggested paper, but they did not discuss appropriately. They cited it after the sentence, "small alpha-dicarbonyls compounds such as glyoxal and methyl glyoxal are important intermediate products of toluene that undergo polymerization to produce low-volatility oligomers", but this paper reports the formation of highly functionalized products such as polyhidroxytoluenes during the toluene oxidation. Please survey previous knowledge of UV/visible absorption spectra for polyhydroxytoluenes (e.g., methylcatechol and methylbenzenetriols), and please consider how polyhydroxytoluenes affect aerosol optical properties to respond the comment. If necessary, please add discussion to the manuscript. Referee #2 suggested that poly hidroxytoluene products are just an example of newly found products. According to knowledge of this referee, other newly found products are highly oxygenated molecules (HOMs) (Molteni et al., 2018, please see the previous reference list of referee #1). In addition to products proposed by Ji et al., it may be useful to discuss whether or not the products found by Molteni et al. affect aerosol optical properties by surveying absorption spectra of organic peroxides.*

Response: Thanks for the suggestions. We have re-read the reference and re-sited the paper in the manuscript: "Ji et al. have reported that toluene oxidation proceeded dominantly via the cresol pathway and formed highly functionalized products such as polyhydroxytoluenes (Ji et al., 2017), which would undergo subsequent reactions with OH to form the precursors including α-carbonyl compounds, organic acids, and other highly oxygenated low-volatility products. Although the peak of 3-methylcatechol, one of the major polyhydroxytoluenes of toluene oxidation products, appeared around 275 nm in UV-Vis spectrum (Ferris et al., 1971), the subsequent products like nitrocatechol and low-volatility oligomers formed by small α-carbonyl compounds might absorb light near ultraviolet and visible. Parenthetically, the reaction rate of 3-methylcatechol with OH radicals was $7.44 \times 10^{-11}$ cm$^3$/molecule·s, and the estimation of atmospheric lifetime was 22 min approximately (Coeur-Tourneur et al., 2010). Small α-carbonyl compounds and organic acids might undergo particle phase reactions, e.g., acid-catalysed aldol condensation reactions under $SO_2$ conditions, plausibly contributing to the observed light absorption (Fu et al., 2009;Fu et al., 2008)." (Line 20-31, page 9).

We also added the discussion of HOMs: "As proved in the study of Moltein et al. (Molteni et al., 2018), highly oxygenated molecules (HOMs) were also important oxidation products of toluene. HOMs that contained a structure of phenyl ring could not absorb light above 300 nm, however, the subsequent HOMs of nitration and ring-opening might absorb UV-Vis light. HOMs were detected in our experiments, especially under high humidity conditions." (Line 20-24, page 8).

Reference:

Ferris, J., Briner, R., and Boyce, C.: Lythraceae alkaloids. IX. Isolation and structure Elucidation of the alkaloids of Lagerstroemia indica, Journal of the American Chemical Society, 93, 2958-2962, 1971.

Coeur-Tourneur, C., Cassez, A., and Wenger, J. C.: Rate coefficients for the gas-phase reaction of hydroxyl radicals with 2-methoxyphenol (guaiacol) and related compounds, The Journal of Physical Chemistry A, 114, 11645-11650, 2010.

Molteni, U., Bianchi, F., Klein, F., El Haddad, I., Frege, C., Rossi, M. J., and Dommen, J.: Formation of highly oxygenated organic molecules from aromatic compounds, Atmospheric Chemistry and Physics, 18, 1909-1921, 2018.

*(2) Line 22, page 2. According to knowledge of this referee, oxides of nitrogen are a mixture of gases that are composed of nitrogen and oxygen and it contains gases other than NO and NO2. The accurate terminology is "nitrogen oxides (NOx)".*

Response: Thanks for the suggestion. We have changed the words to "nitrogen oxides (NO$x$)", "… the effects of nitrogen oxides (NO$x$) on the optical properties of …" (Line 22, page 2).

*(3) Line 1, page 3. According to the format, you may need to remove the initial of first name form the citation, "M. Jaoui et al. (2008)"*

Response: Thanks for the reminding. We have converted the citation to "Jaoui et al. (2008) showed that …" (Line 1, page 3).

*(4) Line 31, page 3. You might have to insert the following sentence written in your response, "unlike NOx analysers equipped with molybdenum converters, HONO was not detected in T200UP NOx analyzer combined with a patented high efficiency potolytic converter." to satisfy a requirement from referee #1.*

Response: Thanks for the suggestion. We have added the sentence in the manuscript, "Parenthetically, unlike NO$x$ analysers equipped with molybdenum converters, HONO was not detected in T200UP NO$x$ Analyser with a patented high efficiency photolytic converter." (Line 16-18, page 4).

*(5) Line 31, page 3. You might have to use the singular for the subject in the sentence, "the concentrations of toluene were measured by a proton transfer reaction quadrupole mass spectrometry…".*

Response: Thanks for the suggestion. We have rewritten the sentence to "The concentration of toluene was measured by a proton transfer reaction quadrupole mass

spectrometry…" (Line 31, page 3).

*(6) Line 28, page 4. The word, "200", should be written in the roman font.*
*''*

Response: Thanks for the suggestion. We have changed the word "200" to the roman font, "…was achieved at *m/z* 200, with an absolute…" (Line 30, page 4).

*(7) Line 38, page 7. A hyphen is missing between "200" and "400".*

Response: Thanks for the suggestion. We have added a hyphen between "200" and "400", "products around *m/z* 200-400 and discussed …" (Line 2, page 8).

*(8) Section 3.2. According to a previous comment by referee #1, the subject, "the complex refractive indexes," in line 2 of page 8 should be "the real part of the complex refractive indexes". All three abbreviations, RI, appearing in the second paragraph of page 8 should be RI(n).*

Response: Thanks for the suggestions. We have changed the "complex refractive indexes" to "the real part of the complex refractive indexes", " the real part of the complex refractive indexes increased from…" (Line 6, page 8) and all the "RI" to "RI(n)" in the second paragraph in Section 3.2 (Line 13-19, page 8). We have also corrected other abbreviations through the particle, e.g., "… overestimate the values of RI(n)." (Line 18, page 9).

*(9)Line 12, page 9. The word, "oligomerization", may be better than "polymerization" according to a previous comment by referee #1.*

Response: Thanks for the suggestion. We have detected the word "polymerization" in the new version.

*(10)Line 34, page 9. You might want to insert "of the WS condition" after the words, "toluene-derived SOA".*

Response: Thanks for the suggestion. We have added words "of the WS conditions" after "toluene-derived SOA", "… RI(n) of toluene-derived SOA of the WS condition were lower than…" (Line 12, page 10).

*(11) Figure S3. The vertical axis title, absorption, will be vague. Does it represent the absorbance based on common logarithm?*

Response: Thanks for the suggestion. As the referee inferred, the word "absorption" in Figure S3 represents the absorbance based on common logarithm. To avoid misunderstanding, we have changed the vertical axis title of Figure S3 to "absorbance".

[Figure]

**Figure S3.** UV-Vis absorption spectrum of toluene SOA under the four different conditions.